

Technical Note: Multiple wavelet coherence for untangling scale-specific
and localized multivariate relationships in geosciences
Wei Hu[2,3] and Bing Cheng Si[1,3]
[1]*College of Hydraulic and Architectural Engineering, Northwest A&F University, Yangling*
*712100, China*
[2]*New Zealand Institute for Plant & Food Research Limited, Private Bag 4704, 8140 Christchurch,*
*New Zealand*
[3]*University of Saskatchewan, Department of Soil Science, Saskatoon, SK S7N 5A8, Canada*
*Correspondence to:* Wei Hu (wei.hu@plantandfood.co.nz) and Bing Cheng Si (bing.si@usask.ca)
**Abstract**
The scale-specific and localized bivariate relationships in geosciences can be
revealed using simple wavelet coherence. The objective of this study is to develop a
multiple wavelet coherence method for examining scale-specific and localized
multivariate relationships. Stationary and non-stationary artificial datasets, generated
with the response variable as the summation of five predictor variables (cosine waves)
with different scales, were used to test the new method. Comparisons were also
conducted using existing multivariate methods including multiple spectral coherence
and multivariate empirical mode decomposition (MEMD). Results show that multiple
spectral coherence is unable to identify localized multivariate relationships and
underestimates the scale-specific multivariate relationships for non-stationary
processes. The MEMD method was able to separate all variables into components at





the same set of scales, revealing scale-specific relationships when combined with
multiple correlation coefficients, but has the same weakness as multiple spectral
coherence. However, multiple wavelet coherences are able to identify scale-specific
and localized multivariate relationships, as they are close to 1 at multiple scales and
locations corresponding to those of predictor variables. Therefore, multiple wavelet
coherence outperforms other common multivariate methods. Multiple wavelet
coherence was applied to a real dataset and revealed the optimal combination of
factors for explaining temporal variation of free water evaporation at Changwu site in
China at multiple scale-location domains. Matlab codes for multiple wavelet
coherence are developed and provided in the supplement.
**1. Introduction**
Geoscience data such as topography, climate, and ocean waves usually present
cyclic patterns, with high-frequency (small-scale) processes being superimposed on
low-frequency (large-scale) processes (Si, 2008). More often than not, geoscience
data is non-stationary, consisting of a variety of frequency regimes that may be
localized in space or time (Torrence and Compo, 1998; Si and Zeleke, 2005; Graf et
al., 2014). The wavelet method is a common tool for detecting multi-scale and
localized features of non-stationary processes in geosciences. Simple wavelet
coherency has been widely used for untangling scale-specific and localized
relationships for non-stationary processes in areas including geophysics (Lakshmi et
al., 2004; Müller et al., 2008), hydrology (Labat et al., 2005; Das and Mohanty, 2008;





Tang and Piechota, 2009; Carey et al., 2013; Graf et al., 2014), soil science (Si and
Zeleke, 2005; Biswas and Si, 2011), meteorology (Torrence and Compo, 1998), and
ecology (Polansky et al., 2010). This method, however, is limited to two variables.
Processes in geosciences are usually complex and may be affected by more than two
environmental factors. A method is needed for analyzing multivariate (>2 variables)
and localized relationships at multiple scales.

49        Several methods have been used for characterizing multivariate relationships. For

example, multiple spectral coherence (MSC) has been used to explore the
scale-specific relationships between soil saturated hydraulic conductivity ($K_s$) and
multiple soil physical properties (Koopmans, 1974; Si, 2008), but requires a stationary
data series which is rare in geosciences. Multivariate empirical mode decomposition
(MEMD), a data-driven method, decomposes each variable into different components
(intrinsic mode functions (IMFs)) with each IMF corresponding to a "common scale"
inherent in multiple variables (Rehman and Mandic, 2010). The MEMD method is
meritorious due to its ability to deal with both non-stationary and nonlinear systems.
The combination of squared multiple correlation coefficient and MEMD ($MCC_{memd}$)
has been used to explore the multivariate control of soil water content or saturated
hydraulic conductivity at multiple scales (Hu and Si, 2013; She et al., 2013, 2015; Hu
et al., 2014). However, the sum of variances from different components typically does
not equal the total variance of the original series, which may result in misleading
$MCC_{memd}$ results. In addition, in geosciences, multivariate relationships are most
likely to change with time or space due to non-stationarity of the processes involved.



However, localized multivariate relationships are not available using any of the
existing multivariate methods. Therefore, it is required to extend the wavelet
coherence from two variables to multiple variables.
An attempt to extend wavelet coherence from two to three variables has been made
by Mihanović et al. (2009). Their method was also applied later in the marine sciences
(Ng and Chan, 2012a, b). Limitations arise when using three variable wavelet
coherence: first, only two predictor variables are considered; second, the two
predictor variables must be orthogonal. Otherwise, extremely high or low (spurious)
coherence ($\gg 1$ or $<0$) may be produced. This spuriousness is inconsistent with the
definition of coherence and may limit the application of this method in geosciences,
where environmental variables are usually cross-correlated. Therefore, a robust
method for calculating MWC, which produces coherence in the closed interval of [0,
1], is needed.
The objective of this paper is to develop an MWC that applies to cases where there
are multiple environmental variables of which may be cross-correlated. This method
is first tested with artificial datasets to demonstrate its advantages over existing
multivariate methods. It is then applied to a temporal series of evaporation ($E$) from
free water surface and meteorological factors at Changwu site in Shaanxi, China.
**2. Theory**
Simple wavelet coherence can be understood as the traditional correlation
coefficient localized in the scale-location domain (Grinsted et al., 2004). Just as





correlation coefficients can be extensions from two variables to multiple (>2)
variables, wavelet coherence between two variables may also be extended to multiple
variables. Similar to simple wavelet coherence, MWC is based on a series of auto-
and cross-wavelet power spectra at different scales and spatial (or temporal) locations
for the response variable and all predictor variables.

91       Following Koopman (1974), a matrix representation of the smoothed auto- and

cross-wavelet power spectra for multiple predictor variables $X$ ( $X = \{X1, X2, \ldots, Xq\}$ )
can be written as
$$\overleftrightarrow{W}^{X,X}(s,\tau) = \begin{bmatrix} \overleftrightarrow{W}^{X1,X1}(s,\tau) & \overleftrightarrow{W}^{X1,X2}(s,\tau) & \cdots & \overleftrightarrow{W}^{X1,Xq}(s,\tau) \\ \overleftrightarrow{W}^{X2,X1}(s,\tau) & \overleftrightarrow{W}^{X2,X2}(s,\tau) & \cdots & \overleftrightarrow{W}^{X2,Xq}(s,\tau) \\ \vdots & \vdots & & \vdots \\ \overleftrightarrow{W}^{Xq,X1}(s,\tau) & \overleftrightarrow{W}^{Xq,X2}(s,\tau) & \cdots & \overleftrightarrow{W}^{Xq,Xq}(s,\tau) \end{bmatrix}, \tag{1}$$

where    $\overleftrightarrow{W}^{Xi,Xj}(s,\tau)$ is the smoothed auto-wavelet power spectra (when $i{=}j$) or
cross-wavelet power spectra (when $i{\neq}j$) at scale $s$ and spatial (or temporal) location
$\tau$ respectively. For the detailed calculation of smoothed auto- and cross-wavelet
power spectra, see Supplement, Sect. S1.

99       The matrix of smoothed cross wavelet power spectra between response variable $Y$

and predictor variables $Xi$ can be defined as
$$\overleftrightarrow{W}^{Y,X}(s,\tau) = \begin{bmatrix} \overleftrightarrow{W}^{Y,X1}(s,\tau) & \overleftrightarrow{W}^{Y,X2}(s,\tau) & \cdots & \overleftrightarrow{W}^{Y,Xq}(s,\tau) \end{bmatrix}, \tag{2}$$

where    $\overleftrightarrow{W}^{Y,Xi}(s,\tau)$  is the smoothed cross-wavelet power spectra between $Y$ and $Xi$ at
scale $s$ and spatial (or temporal) location $\tau$.
The smoothed wavelet power spectrum of response variable $Y$ is $\overleftrightarrow{W}^{Y,Y}(s,\tau)$.
Following Koopmans (1974), the MWC at scale $s$ and location $\tau$, $\rho_m^2(s,\tau)$, can




be written as

$$\rho_m^{\,2}\left(s,\tau\right)=\frac{\overline{W}^{Y,X}\left(s,\tau\right)\overline{W}^{X,X}\left(s,\tau\right)^{-1}\overline{\overline{W}^{Y,X}}\left(s,\tau\right)}{\overline{W}^{Y,Y}\left(s,\tau\right)}. \tag{3}$$

When only one predictor variable (e.g., *X1*) is included in *X*, Eq. (3) is the equation
for simple wavelet coherence, $\rho_s^{\,2}\left(s,\tau\right)$, between two variables (Torrence and
Webster, 1999; Grinsted et al., 2004):

$$\rho_s^{\,2}\left(s,\tau\right)=\frac{\overline{W}^{Y,X1}\left(s,\tau\right)\overline{\overline{W}^{Y,X1}}\left(s,\tau\right)}{\overline{W}^{X1,X1}\left(s,\tau\right)\overline{W}^{Y,Y}\left(s,\tau\right)}. \tag{4}$$

Therefore, simple wavelet coherence is consistent with multiple wavelet coherence
if only one predictor variable is included. In addition, the wavelet phase between a
response variable (*Y*) and a predictor variable (*X1*) is
$\phi\left(s,\tau\right)=\tan^{-1}\left(\text{Im}\left(W^{Y,X1}\left(s,\tau\right)\right)/\text{Re}\left(W^{Y,X1}\left(s,\tau\right)\right)\right),$ (5)
where Im and Re denote the imaginary and real part of $W^{Y,X1}\left(s,\tau\right)$ respectively.
Note that the phase information between a response variable *Y* and multiple predictor
variables *X* cannot be obtained.
Multiple wavelet coherence at the 95% confidence level is calculated using the
Monte Carlo method (Grinsted et al., 2004). Surrogate spatial series (i.e., red noise) of
all variables are generated with a Monte Carlo simulation based on their first-order
autocorrelation coefficient (AR1). The MWC at each scale and location is calculated
using the simulated spatial series. This is repeated an adequate number of times (e.g.,
1000) (Grinsted et al., 2004). At each scale, MWCs at all locations outside the cones
of influence from all simulations are ranked in ascending order. The value at the 95th



percentile represents the 95% confidence level for the MWC at that scale. The Matlab
codes and user manual document for calculating MWC and significance level are
provided in the Supplement (Sect. S2–S4).

## 3. Data and analysis

### 3.1 Artificial data for method test

The method is tested using a stationary and non-stationary artificial dataset
generated following Yan and Gao (2007). The response variable (y for the stationary
case and z for the non-stationary case) encompasses five cosine waves (y1 to y5 for
the stationary case and z1 to z5 for the non-stationary case) with different
dimensionless scales (Fig. 1). For the stationary case, $y1=\cos(2\pi x/4)$, $y2=\cos(2\pi x/8)$,
$y3=\cos(2\pi x/16)$, $y4=\cos(2\pi x/32)$, and $y5=\cos(2\pi x/64)$, where x=0, 1, 2, …, 255.
There is one regular cycle every 4, 8, 16, 32, and 64 locations, representing
dimensionless scales of 4, 8, 16, 32, and 64 for y1, y2, y3, y4, and y5 respectively
(Fig. 1a). The regular cycles make each predictor and response series stationary. For
the non-stationary case, $z1=\cos(500\pi(x/1000)^{0.5})$, $z2=\cos(250\pi(x/1000)^{0.5})$,
$z3=\cos(125\pi(x/1000)^{0.5})$, $z4=\cos(62.5\pi(x/1000)^{0.5})$, and $z5=\cos(31.25\pi(x/1000)^{0.5})$,
where x=0, 1, 2, …, 255. The equation with the square root of the location term
results in the gradual change in frequency (scale), with the greatest dimensionless
scales of 4, 8, 16, 32, and 64 at the right hand side for z1, z2, z3, z4, and z5
respectively (Fig. 1b). The average scales for these predictor variables are 3, 5, 9, 17,
and 32 respectively. The location-varying scales make each predictor and response





variable non-stationary.
For both the stationary and non-stationary series, the variance of the response
variable is 2.5. The predictor variables, each with a variance of 0.5, are orthogonal to
each other, and contribute equally to the total variance of the response variable. The
cosine-like artificial datasets mimic many time series such as seismic signals,
turbulence, air temperature, precipitation, hydrologic fluxes, and the El
Niño-Southern Oscillation. They also mimic spatial series such as ocean waves,
seafloor bathymetry, land surface topography, and soil water content along a
hummocky landscape in geosciences. Therefore, they are representative of a
geoscience data series and are suitable for testing the new method.
Multiple wavelet coherence between the response variable y (or z) and two (y2 and
y4, or z2 and z4) or three (y2, y3, and y4, or z2, z3, and z4) predictor variables were
calculated. The advantage of the artificial data is that the known scale- and localized
features for all variables, and the known relationships between the response and each
predictor variable are exact. By definition, the coherence is 1 at scales corresponding
to that of included predictor variables and 0 at other scales.
To demonstrate the advantages of MWC in dealing with abrupt changes (a type of
transient and localized feature), the second half of the original series of y2 (or z2) or
y4 (or z4) is replaced by 0, and MWC between the response variable and new set of
predictor variables is calculated. We anticipate that the coherence changes from 1 to 0
at the location where the new predictor variable becomes 0.
Predictor variables may not be as regular as that shown in Fig. 1 and may also be





cross-correlated to one another. For these reasons, white noises with a mean of 0 and a
standard deviation of 0.3, 1, and 4 are generated and added to predictor variables of
y2 (or z2) and y4 (or z4). The resulting noised series are termed weakly, moderately,
and highly noised series respectively, and have a correlation coefficient of 0.9, 0.5,
and 0.1 respectively, with their original predictor variable. Multiple wavelet
coherences between the response variable and different predictor variables (original
and noised series) are calculated to demonstrate the performance of MWC when
noised or correlated predictor variables are involved. Only the non-stationary case
will be demonstrated because the performances of MWC for stationary and
non-stationary cases are similar.
The MWC is compared to the MSC (Koopmans, 1974; Si, 2008) and $MCC_{memd}$ (Hu
and Si, 2013). The MSC is calculated based on the calculated auto- and cross- power
spectra using an equation similar to Eq. (3). The detailed introduction of this method
can be found in Si (2008). For the calculation of $MCC_{memd}$, a set of response and
predictor variables form a multivariate data series for MEMD. The MEMD is a data
driven method and has the ability to align "common scales" present within
multivariate data. Please refer to Rehman and Mandic (2010) and Hu and Si (2013)
for the MEMD analysis and the website
(http://www.commsp.ee.ic.ac.uk/~mandic/research/emd.htm) for the related Matlab
codes. The original series of response and predictor variables can be decomposed into
different components (IMFs) with different scales by the MEMD. For IMFs at the
same scale, multiple stepwise regressions are conducted between response and





predictor variables, and the multiple correlation coefficients for each scale-specific
IMF are calculated.

### 3.2 Real data for application

Daily evaporation (*E)* from free water surfaces of E601 evaporation pan (pan
diameter of 61.8 cm) and other meteorological factors (i.e., relative humidity, mean
temperature, sun hours, and wind speed) were collected from January 1, 1979 to
December 31, 2013 at Changwu site in Shaanxi, China. The Changwu site is a
transition area between semi-arid and subhumid climate where water limits
agricultural productivity. Monthly averages of all variables were used in this study
because we are mainly interested in seasonal and inter-annual variability.

## 4.  Results and discussion

### 4.1 MWC with orthogonally predictor variables

For the stationary data, there are two narrow horizontal bands (red color)
representing an MWC value of around 1 at the respective scales of 8 and 32 for all
locations (Fig. 2a). These two bands also correspond to the scales of 8 and 32
respectively, for the two predictor variables. When an additional predictor variable
with the scale of 16 is introduced, a wide band from 6 to 40 appears, signifying that
the MWC equals approximately 1 at all locations at the scales of 8, 16, and 32. As
anticipated, when all five predictor variables with scales ranging from 4 to 64 are
included, coherence values of close to 1 are found in the whole scale-location domain





(data not shown).
The application of MWC to the non-stationary datasets shows that the scales with
significant MWC values gradually increase with the increase in distance. This
increase in the scales is due to the non-stationarity of the variables (Fig. 2b). For
example, when predictor variables of z2 and z4 are included, scales of the two bands
corresponding to MWC around 1 increase from 4 to 8 and from 8 to 32, respectively.
Furthermore, as expected, for only one predictor variable (stationary and
non-stationary), MWC reduces to simple wavelet coherence; there is only one band of
coherence around 1, which corresponds to the scale of that predictor variable (data not
shown). Note that the significant MWC values for both stationary and non-stationary
cases are not exactly 1 at all scales or locations due to the smoothing effect along both
scales and locations. However, the mean MWC values of the significant bands are
very high (i.e., 0.94 – 1.00) and the MWC values at the centre of the significant band
are 1, which corresponds to the exact scale of a predictor variable.
When the point values in the second half of the data series of a predictor variable is
replaced by 0, the MWC in that half is almost 0 at scales corresponding to that
predictor variable (Fig. 3). For the stationary case, when the point values in the
second half of the data series of predictor variable y2 (or y4) is replaced by 0, the
MWC is around 1 at the scale of 8 (or 32) in the first half of the transect and 0 in the
second half (Fig. 3a). Similar results were also found for the non-stationary case (Fig.
3b). This is expected because the constant series of 0 is not correlated to the response
variables at any scale. Much like simple wavelet coherence, the MWC method is able



to detect abrupt changes in the data series and has the advantages of dealing with
localized multivariate relationships.
**4.2 MWC with noised and correlated predictor variables**
When $z2$ and a noised series derived from $z2$ are included as predictor variables,
there is only one band of coherence close to 1 at scales corresponding to $z2$,
irrespective of the correlation between $z2$ and a noised series of $z2$ (Fig. 4a). When $z2$
and a noised series of $z4$ are included as predictor variables, the coherence depends on
the degree of the noise (Fig. 4b). For weakly noised series, there are two bands of
coherence of around 1 corresponding to the scales of $z2$ and $z4$ respectively. The
PASC is 23%, which equals that of when $z2$ and $z4$ are included. With the increase of
noise, the coherence and corresponding PASC at the scales corresponding to $z4$
decrease. When $z2$ and a strongly noised series of $z4$ are considered, the band of
coherence around 1 at scales corresponding to $z4$ disappears.
The inclusion of a third noised $z4$ variable substantially increases the area with high
coherence (in red) as compared to the case when only $z2$ and $z4$ are included (Fig. 4c).
This indicates that MWC will increase with the increase in the number of predictor
variables, with the highest coherence less or equal to 1, irrespective of the number of
predictor variables. However, the area of significant coherence may not necessarily
increase (Ng and Chan, 2012a). In fact, the PASC values for three predictor variables
(19-20%) are lower than for only two predictor variables (23%). This indicates that, in
this case, two predictor variables are better than three in terms of explaining the





variations of the response variable. This is because the variance of the response
variable explained by the noised variable is already accounted for by other variables.
Therefore, only an additional variable that can independently explain a fair amount of
variance could contribute significantly to explaining variations of a response variable
(Fig. 4b). This can also explain why there is only one band of coherence around 1 at
scales corresponding to $z2$, when $z2$ and a noised series of $z2$ are included (Fig. 4a).
This information is helpful in choosing predictor variables for developing
scale-specific predictions, especially when predictor variables are correlated.
**4.3 Comparison with other multivariate methods**
4.3.1 MSC
The MSC as a function of scale is shown in Fig. 5a. For the stationary case, when
$y2$ and $y4$ are included as predictor variables, there are two plateaus centered at the
scales of 8 and 28 representing a coherence of 1. As expected, when an additional
predictor variable $y3$ is added, the corresponding scale of 16 also shows coherence of
1. The MSC produces similar scale-specific relationships as MWC does for a
stationary dataset except that the centered scale (i.e., 28) with coherence of 1 deviates
from the expected value (i.e., 32) for predictor variable $y4$. For the non-stationary
case, however, the MSC is much lower than 1 for the predictor variables of $z2$ and $z4$;
MSC of 1 is present only at the scale of 8 when an additional predictor variable $z3$ is
added. Obviously, the MSC underestimates the multivariate relationships and is not
suitable to non-stationary processes (Si, 2008) due to its inability to deal with



localized features. The MSC at a specific scale provides the average of multivariate
relationships across all locations. Because the scale of a predictor variable changes
with location for the non-stationary case, the MSC deviates greatly from 1.
The inability of the MSC to deal with localized features is demonstrated further by
the decrease of MSC at scales when the second half of the included predictor variable
series are replaced by 0 for both the stationary and non-stationary series (Fig. 5b). For
example, when the second half of the y4 series is replaced by 0 for the stationary case,
the MSC at scales around 32 decreases from 1 to 0.52. Although the MSC can detect
the decrease of coherence at the scales corresponding to the 0 values throughout the
second half of the series, the exact locations for the decrease cannot be identified. In
fact, the coherence decreases only in the second half of the series, and does not
change in the first half of the series. The location for the decrease can be easily
identified by the MWC, but not by MSC.
4.3.2 MCC$_{memd}$
Five intrinsic mode functions (IMFs) with non-negligible variance are obtained for
multivariate data series. While the obtained scales for the response variable y are in
agreement with the true scales for the stationary case, the obtained scales (i.e., 3, 6, 11,
21, and 43) for the response variable z deviate slightly from the average scales for the
non-stationary case. For the response variable, the contribution of IMFs to the total
variance generally decreases (20% to 13% for stationary and 27% to 11% for
non-stationary) from IMF1 to IMF5, which disagrees with the fact that each scale





contributes equally (i.e., 20%) to the total variance. In addition, the sum of variances
over all IMFs for each variable is less than 100% (ranging from 84% to 93%),
indicating that MEMD cannot capture all the variances. For the detailed results of
MEMD, see Supplement, Sect. S5.
The $MCC_{memd}$ as a function of scale is shown in Fig. 6a. For the stationary case,
when predictor variables of y2 and y4 are included, the $MCC_{memd}$ values are 0.98 and
0.93 respectively, at scales corresponding to that of y2 and y4. When a predictor
variable of y3 is included, the $MCC_{memd}$ values are 1.00, 1.00, and 0.96 respectively,
at scales corresponding to that of y2, y3, and y4. For the non-stationary case, the
corresponding $MCC_{memd}$ values are 0.80 and 0.85 for the two predictor variable case,
and 0.95, 0.99, and 0.91 respectively, for the case of three predictor variables.
Therefore, the $MCC_{memd}$ can be used to determine the scale-specific multivariate
relationships. Similar to MSC, however, the $MCC_{memd}$ underestimates the multivariate
relationships, especially for the non-stationary case with less predictor variables. On
the contrary, the $MCC_{memd}$ can also overestimate the multivariate relationships. For
example, when only predictor variables corresponding to scales of 8, 16, and 32 are
considered, the $MCC_{memd}$ value for the stationary case is 0.47 at the scale of 64, which
deviates much from the expected $MCC_{memd}$ value of 0 (Fig. 6a). The possible
underestimation and overestimation by the $MCC_{memd}$ may come from the
decomposition errors inherent in the MEMD algorithm (Rehman and Mandic, 2010).
Similar to MSC, the localized multivariate relationships cannot be obtained from
$MCC_{memd}$. This can be better explained by the decrease of $MCC_{memd}$ when half of the



series of the predictor variables are replaced by 0 (Fig. 6b). For the stationary case,
the $MCC_{memd}$ values at the scales corresponding to y2 (or y4) decrease from 0.98 to
0.49 and from 0.93 to 0.62 when the second half of the y2 (or y4) series are replaced
by 0.
As explained above, the MWC has advantages in untangling localized multivariate
relationships as compared to the common multivariate methods. It is important to
reveal the multivariate relationships, which vary with time or space that are associated
with different processes. For example, discharge usually happens on knolls, while
recharge usually happens in neighboring depressions (Gates et al., 2011). Therefore,
the controlling factors of soil water storage may vary with the land element
characteristics of a location. For example, local controls may be more important on
knolls, while non-local controls may be more important in depressions (Grayson et al.,
1997). In a temporal domain, vegetation transpiration contributes more to the
evapotranspiration in the growing seasons, which may result in the changes of
environmental factors explaining temporal variations of evapotranspiration in
different seasons.
**4.4 Application of the MWC**
Each meteorological factor was significantly correlated to the $E$, but the dominant
factors explaining variation in $E$ differed with scale. For example, the relative
humidity was the dominating factor at small (2–8 months) and large (>32 months)
scales, while temperature was the dominating factor at the medium (8–32 months)



scales. Overall, the relative humidity corresponded to the greatest mean MWC (0.62)
and PASC value (40%) at multiple scale-location domains. For the detailed
relationships between $E$ and each factor, see Supplement, Sect. S6.
The MWC analysis shows that the combination of relative humidity and mean
temperature produced the greatest mean MWC (0.82) and PASC (49%) among all
two-factor cases, indicating that they are the best to explain variations in $E$ at multiple
scale-location domains (Fig. 7a). However, adding an additional factor such as sun
hours, which was the best among all three-factor cases, increased the average
coherence (0.91), but slightly decreased the PASC to 48% (Fig. 7b). This indicated
that sun hours was not significantly different from red noise in explaining additional
variation in $E$. Similar results were found when the wind speed was added. The reason
for this was that most areas with significant coherence between $E$ and sun hours or
wind speed, were a subset of areas with significant coherence between $E$ and relative
humidity or mean temperature (see Supplement, Sect. S3). Therefore, relative
humidity and mean temperature were adequate to explain the temporal variation of $E$
at various scales at this site. This is consistent with Li et al. (2012), who indicate that
relative humidity and mean temperature are the two main contributors to the temporal
change of potential evapotranspiration on the Chinese Loess Plateau.
**5. Conclusions**
Multiple wavelet coherence is developed to determine scale-specific and localized
multivariate relationships in geosciences. The new method is tested and compared



with exiting multivariate methods using an artificial dataset. The new method can be
used to determine the proportion of the variance of a response variable that is
explained by predictor variables at a specific scale and location (spatially or
temporally). As compared with simple wavelet coherence, more variation may be
explained at multiple scale-location domains by the MWC. Including more variables
is only beneficial if the variables are not strongly cross-correlated and can
independently explain a fair amount of variability in a response variable. Therefore,
the best combinations of variables that explain multivariate spatial or temporal
variability at multiple scales can be determined. This is important for optimizing
variables for developing scale-specific prediction. The MSC and $MCC_{memd}$ can
determine multivariate relationships at multiple scales, but localized multivariate
relationships are not available and both MSC and $MCC_{memd}$ are likely to
underestimate the degree of multivariate relationships for non-stationary processes. In
addition, the performance of $MCC_{memd}$ relies on the performance of MEMD, which
needs further development. Application of the MWC into the real dataset indicates
that the combination of relative humidity and mean temperature are the optimal
factors to explain temporal variation of $E$ at the Changwu site in China.
In summary, multiple wavelet coherence has advantages over existing multivariate
methods, and provides an effective vehicle for untangling complex spatial or temporal
variability for multiple controlling factors at multiple scales and locations. It may also
be used as a data-driven tool for modeling and predicting various processes in the area
of geosciences such as precipitation, drought, soil water dynamics, stream flow, and



atmospheric circulation.
**Acknowledgements**
The Matlab codes for calculating multiple wavelet coherence are developed based on
the codes provided by A. Grinsted
(http://noc.ac.uk/using-science/crosswavelet-wavelet-coherence) and, together with
user manual, are available in the Supplement (Sect. S2-S4). The project was partially
funded by the Natural Sciences and Engineering Research Council of Canada
(NSERC) and Agriculture Development Fund of Saskatchewan.

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

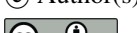



**Figure captions**
**Figure 1**. (a) Stationary and (b) non-stationary series of response variables (y for
stationary and z for non-stationary case) encompassing five cosine waves (y1 to y5
for stationary and z1 to z5 for non-stationary case) with different dimensionless
scales.
**Figure 2.** Multiple wavelet coherence (a) between response variable y and predictor
variables y2 and y4; (b) between response y and predictors y2, y3, and y4; (c)
between response z and predictors z2 and z4; and (d) between response z and
predictors z2, z3, and z4. The artificial data series (y) encompasses five cosine waves
(y1, y2, y3, y4, and y5) with different scales for the stationary case, and the artificial
data series (z) encompasses five cosine waves (z1, z2, z3, z4, and z5) with different
scales for the non-stationary case. The predictor variables, connected by a hyphen, are
shown in the top right corner of each subplot. Thin solid lines demarcate the cones of
influence, and thick solid lines show the 95% confidence levels.
**Figure 3.** Multiple wavelet coherence (a) between y and y2h0 and y4; (b) between y
and y2 and y4h0; (c) between z and z2h0 and z4; and (d) between z and z2 and z4h0.
The artificial data series (y) encompasses five cosine waves (y1, y2, y3, y4, and y5)
with different scales for the stationary case and the artificial data series (z)
encompasses five cosine waves (z1, z2, z3, z4, and z5) with different scales for the
non-stationary case. The variables y2h0 (or z2h0) and y4h0 (or z4h0) refer to the new
series of y2 (or z2) and y4 (or z4), in which the second half is replaced by 0. The

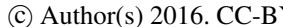


predictor variables, connected by a hyphen, are shown in the top right corner of each
subplot. Thin solid lines demarcate the cones of influence and thick solid lines show
the 95% confidence levels.
**Figure 4.** Multiple wavelet coherence of an artificial data series (z) encompassing five
cosine waves (z1, z2, z3, z4, and z5) with different scales and (a) z2 and noised z2, (b)
z2 and noised z4, and (c) z2, z4, and noised z4 for the non-stationary case. The
predictor variables are connected by a hyphen and shown in the top right corner of
each subplot. z2wn (z4wn), z2mn (z4mn), and z2sn (z4sn) indicate weakly,
moderately, and strongly noised z2 (z4) series respectively. Weakly, moderately, and
strongly noised series are correlated with original series, having correlation
coefficients of 0.9, 0.5, and 0.1 respectively. Thin solid lines demarcate the cones of
influence and thick solid lines show the 95% confidence levels.
**Figure 5.** Multiple spectral coherence (MSC) of an artificial data series (y or z)
encompassing five cosine waves (y1 to y5; or z1 to z5) with different scales and (a)
two (y2 and y4; or z2 and z4) or three (y2, y3, and y4; or z2, z3, and z4) data series,
and (b) two (y2 and y4; or z2 and z4) data series when the second half of one data
series is replaced by 0. The variables y2h0 (or z2h0) and y4h0 (or z4h0) refer to the
new series of y2 (or z2) and y4 (or z4) in which the second half is replaced by 0.
**Figure 6.** Multiple correlation coefficient between multivariate empirical mode
decomposition ($MCC_{memd}$) of an artificial series (y or z) and (a) two (y2 and y4; or z2
and z4) or three (y2, y3, and y4; or z2, z3, and z4) data series, and (b) two (y2 and y4;
or z2 and z4) data series when the second half of one data series is replaced by 0. The



variables y2h0 (or z2h0) and y4h0 (or z4h0) refer to the new series of y2 (or z2) and
y4 (or z4) in which the second half is replaced by 0.
**Figure 7.** Multiple wavelet coherence between evaporation ($E$) from water surfaces
and meteorological factors ((a) relative humidity and mean temperature and (b)
relative humidity, mean temperature, and sun hours) at Changwu site in Shaanxi,
China. Thin solid lines demarcate the cones of influence, and thick solid lines show
the 95% confidence level.



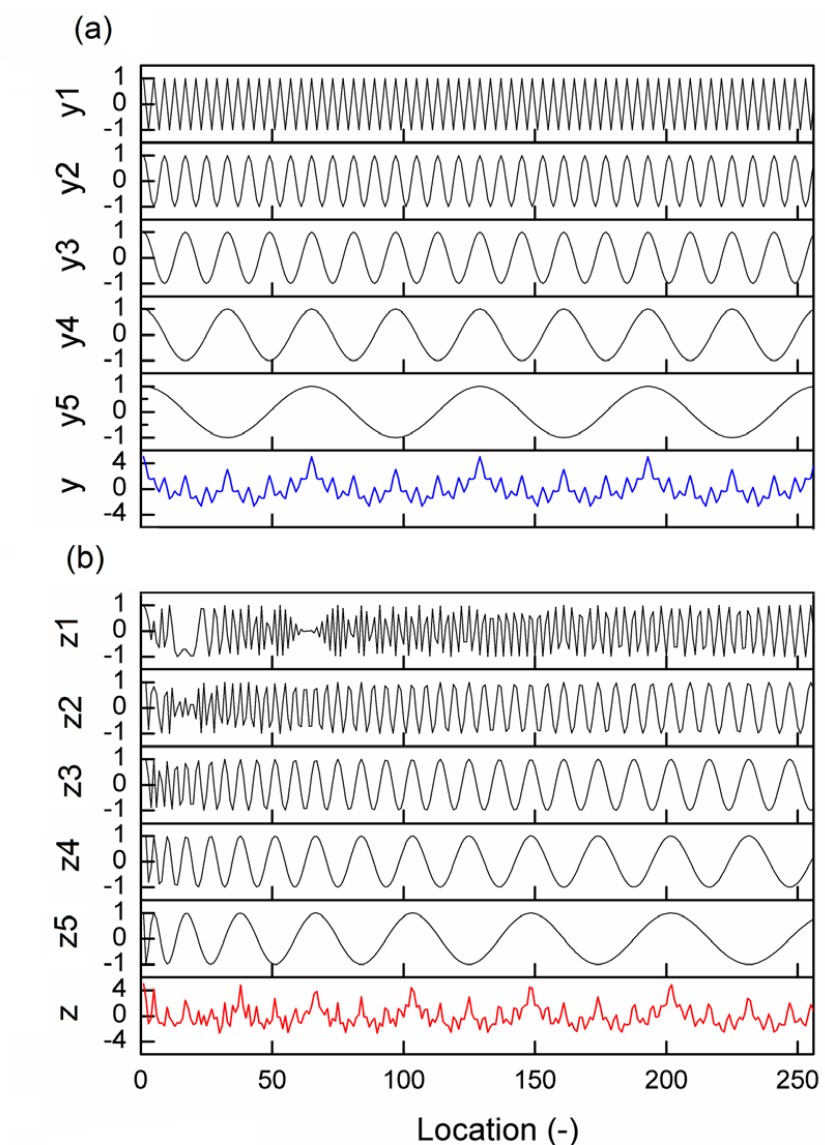

**Figure 1**. (a) Stationary and (b) non-stationary series of response variables (y for stationary and z for non-stationary case) encompassing five cosine waves (y1 to y5 for stationary and z1 to z5 for non-stationary case) with different dimensionless scales.

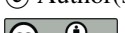



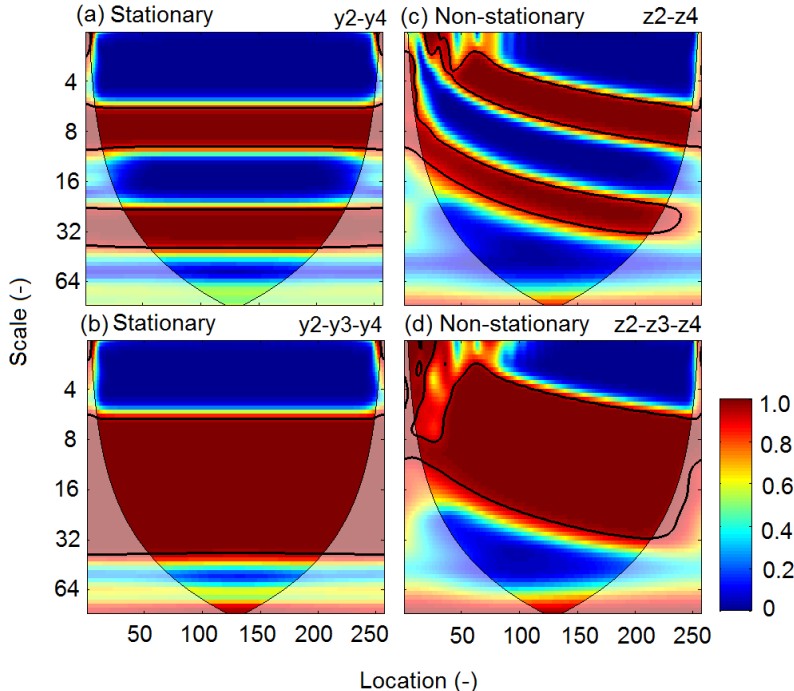

**Figure 2.** Multiple wavelet coherence (a) between response variable y and predictor variables y2 and y4; (b) between response y and predictors y2, y3, and y4; (c) between response z and predictors z2 and z4; and (d) between response z and predictors z2, z3, and z4. The artificial data series (y) encompasses five cosine waves (y1, y2, y3, y4, and y5) with different scales for the stationary case, and the artificial data series (z) encompasses five cosine waves (z1, z2, z3, z4, and z5) with different scales for the non-stationary case. The predictor variables, connected by a hyphen, are shown in the top right corner of each subplot. Thin solid lines demarcate the cones of influence, and thick solid lines show the 95% confidence levels.





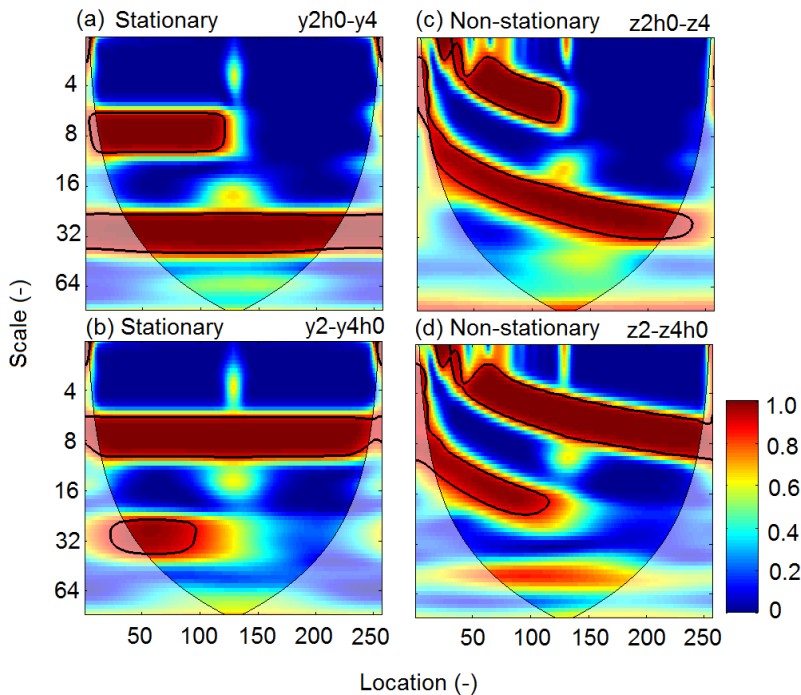

**Figure 3.** Multiple wavelet coherence (a) between y and y2h0 and y4; (b) between y and y2 and y4h0; (c) between z and z2h0 and z4; and (d) between z and z2 and z4h0. The artificial data series (y) encompasses five cosine waves (y1, y2, y3, y4, and y5) with different scales for the stationary case and the artificial data series (z) encompasses five cosine waves (z1, z2, z3, z4, and z5) with different scales for the non-stationary case. The variables y2h0 (or z2h0) and y4h0 (or z4h0) refer to the new series of y2 (or z2) and y4 (or z4), in which the second half is replaced by 0. The predictor variables, connected by a hyphen, are shown in the top right corner of each subplot. Thin solid lines demarcate the cones of influence and thick solid lines show the 95% confidence levels.





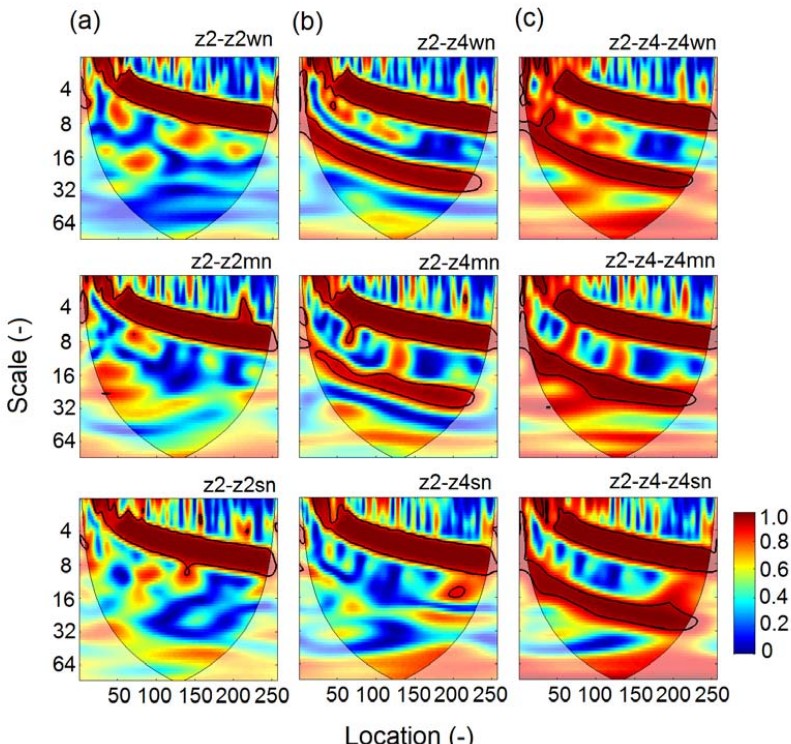

**Figure 4.** Multiple wavelet coherence of an artificial data series (z) encompassing five cosine waves (z1, z2, z3, z4, and z5) with different scales and (a) z2 and noised z2, (b) z2 and noised z4, and (c) z2, z4, and noised z4 for the non-stationary case. The predictor variables are connected by a hyphen and shown in the top right corner of each subplot. z2wn (z4wn), z2mn (z4mn), and z2sn (z4sn) indicate weakly, moderately, and strongly noised z2 (z4) series, respectively. Weakly, moderately, and strongly noised series are correlated with original series, having with correlation coefficients of 0.9, 0.5, and 0.1, respectively. Thin solid lines demarcate the cones of influence and thick solid lines show the 95% confidence levels.





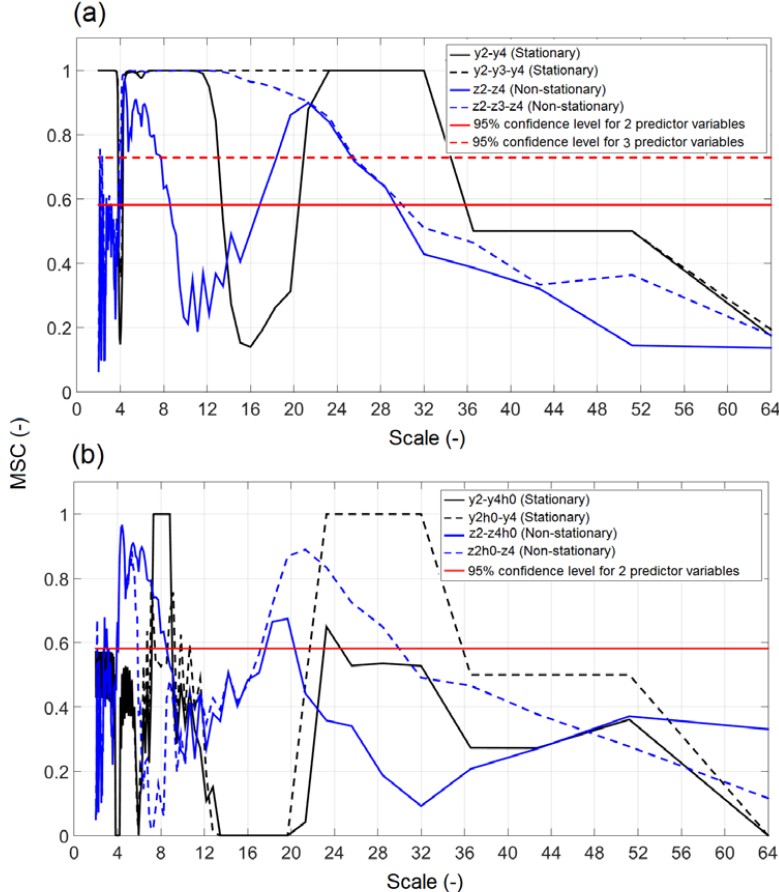

**Figure 5.** Multiple spectral coherence (MSC) of an artificial data series (y or z) encompassing five cosine waves (y1 to y5; or z1 to z5) with different scales and (a) two (y2 and y4; or z2 and z4) or three (y2, y3, and y4; or z2, z3, and z4) data series, and (b) two (y2 and y4; or z2 and z4) data series when the second half of one data series is replaced by 0. The variables y2h0 (or z2h0) and y4h0 (or z4h0) refer to the new series of y2 (or z2) and y4 (or z4) in which the second half is replaced by 0.





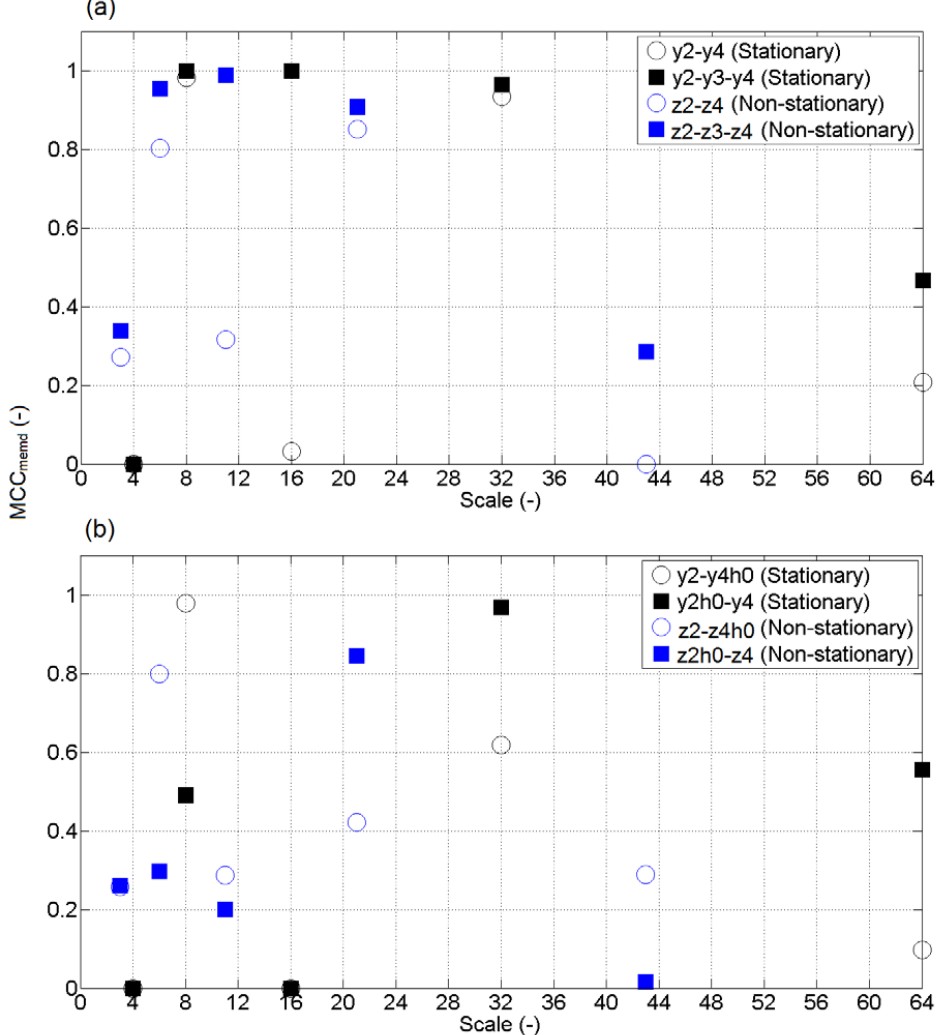

**Figure 6.** Multiple correlation coefficient between multivariate empirical mode decomposition ($MCC_{memd}$) of an artificial series (y or z) and (a) two (y2 and y4; or z2 and z4) or three (y2, y3, and y4; or z2, z3, and z4) data series, and (b) two (y2 and y4; or z2 and z4) data series when the second half of one data series is replaced by 0. The variables y2h0 (or z2h0) and y4h0 (or z4h0) refer to the new series of y2 (or z2) and y4 (or z4) in which the second half is replaced by 0.



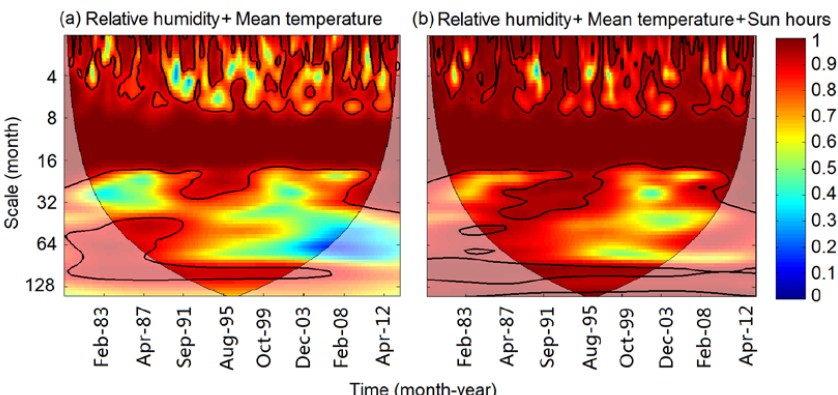

**Figure 7.** Multiple wavelet coherence between evaporation (E) from water surfaces and meteorological factors ((a) relative humidity and mean temperature and, (b) relative humidity, mean temperature, and sun hours) at Changwu site in Shaanxi, China. Thin solid lines demarcate the cones of influence, and thick solid lines show the 95% confidence level.