# Peer review of "Technical Note: Multiple wavelet coherence for untangling scale-specific"

_Hydrology and Earth System Sciences, 2016_

## Referee Comment (RC1) · Anonymous Referee #1 · 29 May 2016

The manuscript of Multiple wavelet coherence by Hu and Si presented an important topic. In characterizing scale specific variations, wavelet coherence has been used in many field but was restricted to only two variables. Presentation of wavelet coherence produces a step forward on the methodological development aspect. The method will support a lot of different fields including soil science and hydrology. The scientific content is suitable for the journal and the readers of this journal will be interested in this topic. Therefore, my suggestion is for acceptance of the manuscript with some minor corrections such as English, which could be improved. Another thing, authors used the artificial series to compare with other multi-variate analysis. Just wondering, how will you confirm about you claimed superior information of the new method compare

to other methods. I mean to say, how will you say that this variations, what is shown by other methods are also showing the right information. The variations showing here could be spurious as identified by different methods.

---

## Referee Comment (RC2) · Anonymous Referee #2 · 22 Jun 2016

**General Comments**

The multiple wavelet coherence methodology presented in the manuscript by Hu and Si represents an important contribution to wavelet analysis. In particular, Hu and Si build upon the previous work of Ng and Chan (2012) to extend multiple wavelet coherence to case of more than two predictor variables. The authors further demonstrate that the new multiple wavelet coherence methodology is better suited for situations where the predictor variables are cross-correlated. The problems with the traditional formulation are clearly stated and consistent with the objective of the paper proposed in the introduction section. Theoretical examples were also presented to highlight the advantages of the new methodology relative to existing ones. I their recommend that

the manuscript be accepted after the substantial correction of grammatical errors and the consideration of more specific comments presented below.

**Specific comments**

The conclusion section simply summarizes the results of the paper. The authors could consider expanding the conclusion section into a discussion section to comment on limitations of the method. After all, wavelet analysis, while useful, is not a scientific panacea. More specifically, the inclusion of more predictor variables may result in the statistical significance threshold at a particular wavelet scale and time to approach unity, which would impose a limit on how much statistical information can be gained. This phenomenon occurs with the traditional multiple wavelet coherence formulation, where the threshold for 5% significance, for example, is higher than that for bivariate wavelet coherence at a given wavelet scale.

The author may also consider discussing at least briefly the problem of simultaneously testing multiple statistical hypothesis, as discussed in Maraun and Kurths (2004), Maraun et al. (2007), Schulte et al. (2015), and Schulte (2016). Multiple-testing problem is a major problem in wavelet analysis and therefore merits consideration in a discussion section. Presenting clearly the methodological limitations will better guide the likely interdisciplinary readership in making decisions regarding what analysis tools to implement.

Throughout the manuscript, the authors mention how geoscience data are often nonstationary. Perhaps the term is used too loosely in some instances and is sometimes inconsistent with the strict time series analysis definition. Even white and red-noise processes contain time and scale-localized features in wavelet space, even though their respective statistics are stationary at all orders. Time- and scale-localized features are evident in the wavelet power spectrum of say, the North Atlantic Oscillation (NAO), even though the statistics of the NAO are consistent with a first-order Markov process (Feldstein, 2000). Therefore, in some instances, I recommend changing the word "non-
stationary" to "transient" or "transitory".

Some Technical Corrections

Page 2 Line 3536. Change "geoscience data is" to "geoscience data are".

Page 2 Line 39. Is it better to say bivariate wavelet coherency rather than "simple wavelet coherency" Page 5, Line 97. Add comma before "respectively".

Page 9, Line 169-171. The sentence can be slightly simplified by changing "white noise with a mean of 0" to "zero-mean white noise". Perhaps it is redundant to write that the white noise processes were generated. Authors could consider just saying that white noise was added to the predictor variables.

Page 9, Lines 171-173. The sentence "The resulting noised series are termed weakly, moderately, 172 and highly noised series respectively, and have a correlation coefficient of 0.9, 0.5, 173 and 0.1 respectively, with their original predictor variable" needs to be rewritten and simplified. Consider breaking the sentence into two separate sentences.

The authors should carefully check for grammatical errors and make similar changes throughout the manuscript.

References

Feldstein SB (2000) The timescale, power spectra, and climate noise properties of teleconnection patterns. J Clim 13:4430–4440. doi:10.1175/1520-0442(2000)0132.0.CO;2 Maraun, D. and Kurths, J.: Cross wavelet analysis: significance testingand pitfalls, Nonlin. Processes Geophys., 11, 505–514, 2004.

Maraun, D., Kurths, J., and Holschneider, M.: Nonstationary Gaussian processes in wavelet domain: synthesis, estimation, and significance testing, Phys. Rev. E, 75, doi:10.1103/PhysRevE.75.016707, 2007.

Ng, E. K. W. and Chan, J. C. L.: Geophysical applications of partial waveletcoher-

**HESSD**
ence and multiple wavelet coherence, J. Atmos. Ocean. Tech., 29, 1845–1853, doi: 10.1175/JTECH-D-12-00056.1, 2012.

Schulte, J. A., Duffy, C., and Najjar, R. G.: Geometric and topological approaches to significance testing in wavelet analysis, Nonlin.Processes Geophys., 22, 139–156, doi:10.5194/npg-22-139-2015, 2015.

Schulte, J. A.: Cumulative areawise testing in wavelet analysis and its application to geophysical time series, Nonlin. Processes Geophys., 23, 45-57, doi:10.5194/npg-23-45-2016, 2016.

---

## Author Comment (AC1) · 25 Jun 2016

The manuscript of Multiple wavelet coherence by Hu and Si presented an important topic. In characterizing scale specific variations, wavelet coherence has been used in many field but was restricted to only two variables. Presentation of wavelet coherence produces a step forward on the methodological development aspect. The method will support a lot of different fields including soil science and hydrology. The scientific content is suitable for the journal and the readers of this journal will be interested in this topic. Therefore, my suggestion is for acceptance of the manuscript with some minor corrections such as English, which could be improved. Another thing, authors used the artificial series to compare with other multi-variate analysis. Just wondering, how will you confirm about you claimed superior information of the new method compare to other methods. I mean to say, how will you say that this variations, what is shown by other methods are also showing the right information. The variations showing here could be spurious as identified by different methods.

Response:

Thank you for the positive comments.

In terms of language, we have tried our best to correct it. We will ask an English editing company check the language again if we will be given a chance for revision.

We are not very sure we understand your second comment, but we will try to explain a bit here. The two existing methods (i.e., multiple spectral coherence and multivariate empirical mode decomposition) are widely used for spatial or temporal series analysis in different disciplines. Actually we have known that these two methods cannot deal with localized relationships between variables. Therefore, the advantages of the new method over these two methods is demonstrated mainly in terms of relationships between response and predictor variables at various scales of the response variable. The reason for using the artificial data is that the major features (e.g., scale) are known. Then, the superiority of the new method over these two methods can be assessed by whether the known major features of the artificial data are demonstrated by these methods. Our results clearly show that localized multivariate relationships are not available by the two existing methods and both methods are likely to underestimate the degree of multivariate relationships for non-stationary processes. Because the cosine-like artificial datasets mimic many time series and spatial series in geosciences. Therefore, we conclude that the new method is superior.

All above mentioned information can be found in the attached revised copy. Please refer to them at Lines 84-86, 156-162, 188-191, and 384-387.

Please also note the supplement to this comment:

http://www.hydrol-earth-syst-sci-discuss.net/hess-2016-154/hess-2016-154-AC1-supplement.pdf

[Figure]

**Supplement:**

[revised manuscript text omitted]
} \overline{\overleftrightarrow{W}}^{X1,X1}(s,\tau) & \overline{\overleftrightarrow{W}}^{X1,X2}(s,\tau) & \cdots & \overline{\overleftrightarrow{W}}^{X1,Xq}(s,\tau) \\ \overline{\overleftrightarrow{W}}^{X2,X1}(s,\tau) & \overline{\overleftrightarrow{W}}^{X2,X2}(s,\tau) & \cdots & \overline{\overleftrightarrow{W}}^{X2,Xq}(s,\tau) \\ \vdots & \vdots & & \vdots \\ \overline{\overleftrightarrow{W}}^{Xq,X1}(s,\tau) & \overline{\overleftrightarrow{W}}^{Xq,X2}(s,\tau) & \cdots & \overline{\overleftrightarrow{W}}^{Xq,Xq}(s,\tau) \end{bmatrix}, \tag{1}$$

where $\overleftrightarrow{W}^{Xi,Xj}(s,\tau)$ is the smoothed auto-wavelet power spectra (when $i{=}j$) or cross-wavelet power spectra (when $i{\neq}j$) at scale $s$ and spatial (or temporal) location

$\tau$, respectively. For the detailed calculation of smoothed auto- and cross-wavelet power spectra, see Supplement, Sect. S1.

The matrix of smoothed cross wavelet power spectra between response variable $Y$

and predictor variables $Xi$ can be defined as

$$\overleftrightarrow{W}^{Y,X}(s,\tau) = \left[ \overleftrightarrow{W}^{Y,X1}(s,\tau) \ \overleftrightarrow{W}^{Y,X2}(s,\tau) \ \cdots \ \overleftrightarrow{W}^{Y,Xq}(s,\tau) \right], \tag{2}$$

Field Code Changed where $\overleftrightarrow{W}^{Y,Xi}(s,\tau)$ is the smoothed cross-wavelet power spectra between $Y$ and $Xi$ at scale $s$ and spatial (or temporal) location $\tau$.

The smoothed wavelet power spectrum of response variable $Y$ is $\overleftrightarrow{W}^{Y,Y}(s,\tau)$.

Following Koopmans (1974), the MWC at scale $s$ and location $\tau$, $\rho_m^2(s,\tau)$, can be written as

$$\rho_m^2(s,\tau) = \frac{\overleftrightarrow{W}^{Y,X}(s,\tau)\overleftrightarrow{W}^{X,X}(s,\tau)^{-1}\overline{\overleftrightarrow{W}^{Y,X}(s,\tau)}}{\overleftrightarrow{W}^{Y,Y}(s,\tau)}. \tag{3}$$

When only one predictor variable (e.g., $X1$) is included in $X$, Eq. (3) is the equation for bivariate<s>simple</s> wavelet coherence, $\rho_b^2(s,\tau)$ <s>$\rho_s^2(s,\tau)$</s> , <s>between two</s>

Field Code Changed

[revised manuscript text omitted]

---

## Author Comment (AC2) · 25 Jun 2016

General Comments The multiple wavelet coherence methodology presented in the manuscript by Hu and Si represents an important contribution to wavelet analysis. In particular, Hu and Si build upon the previous work of Ng and Chan (2012) to extend multiple wavelet coherence to case of more than two predictor variables. The authors further demonstrate that the new multiple wavelet coherence methodology is better suited for situations where the predictor variables are cross-correlated. The problems with the traditional formulation are clearly stated and consistent with the objective of the paper proposed in the introduction section. Theoretical examples were also presented to highlight the advantages of the new methodology relative to existing ones. I their

recommend that the manuscript be accepted after the substantial correction of grammatical errors and the consideration of more specific comments presented below.

Response:

Thank you for the positive comments.

Specific comments The conclusion section simply summarizes the results of the paper. The authors could consider expanding the conclusion section into a discussion section to comment on limitations of the method. After all, wavelet analysis, while useful, is not a scientific panacea. More specifically, the inclusion of more predictor variables may result in the statistical significance threshold at a particular wavelet scale and time to approach unity, which would impose a limit on how much statistical information can be gained. This phenomenon occurs with the traditional multiple wavelet coherence formulation, where the threshold for 5

Response: We agree with you that one of the limitation is that the critical values increase with the number of predictor variables. This is also why the percentage area of significant coherence (PASC) for three predictor variables (z2, z4, and noised z4) are even lower than for only two predictor variables (z2 and z4) when the third predictor variable (noised z4) is not statistically significant to explain the variation of the response variable. Please see Lines 265-266 in the attached revision. We put this limitation in the conclusion part as " Theoretically, any number of predictor variables can be included in the multiple wavelet analysis. However, the statistical significance threshold usually increases with the number of the predictor variables (Grinsted et al., 2004; Ng and Chan, 2012a), and inclusion of too many predictor variables may result in the statistical significance threshold at particular wavelet scales (e.g., the lowest and largest scales) to approach unity. This would restrict the availability of statistical information." (Lines391-397 in the attached revision).

The author may also consider discussing at least briefly the problem of simultaneously testing multiple statistical hypothesis, as discussed in Maraun and Kurths (2004),

Maraun et al. (2007), Schulte et al. (2015), and Schulte (2016). Multiple-testing problem is a major problem in wavelet analysis and therefore merits consideration in a discussion section. Presenting clearly the methodological limitations will better guide the likely interdisciplinary readership in making decisions regarding what analysis tools to implement.

Response: The multiple-testing problem has been briefly discussed in the conclusion part. "In addition, similar to bivariate wavelet analysis, the new method also suffers from the multiple-testing problem (Maraun and Kurths, 2004; Maraun et al., 2007; Schulte et al., 2015; Schulte, 2016). Therefore, a more robust statistical significance testing method may be beneficial to the new method." (Lines397-400 in the attached revision).

Throughout the manuscript, the authors mention how geoscience data are often nonstationary. Perhaps the term is used too loosely in some instances and is sometimes inconsistent with the strict time series analysis definition. Even white and red-noise processes contain time and scale-localized features in wavelet space, even though theirrespectivestatisticsarestationaryatallorders. Time-andscale-localizedfeatures are evident in the wavelet power spectrum of say, the North Atlantic Oscillation (NAO), even though the statistics of the NAO are consistent with a first-order Markov process (Feldstein,2000). Therefore,insomeinstances,Irecommendchangingtheword"nonstationary" to "transient" or "transitory".

Response: We agree. In the introduction, we made this more clear as " More often than not, geoscience data are transient, consisting of a variety of frequency regimes that may be localized in space or time (Torrence and Compo, 1998; Si and Zeleke, 2005; Graf et al., 2014). The transient characteristics exists widely in non-stationary but also sometimes in stationary processes (Feldstein, 2000)." (Lines35-39 in the attached revision). At many instances, we changed the "non-stationary" to "transient" when suitable, such as Line 41, 59, 67 in the attached revision.

[Figure]

Some Technical Corrections Page 2 Line 3536. Change "geoscience data is" to "geoscience data are".

Response: Yes, done at L36.

Page 2 Line 39. Is it better to say bivariate wavelet coherency rather than "simple wavelet coherency"

Response: Yes, we changed all throughout the paper.

Page 5, Line 97. Add comma before "respectively".

Response: Yes, we did throughout the paper.

Page9,Line169-171. The sentence can be slightly simplified by changing" white noise with a mean of 0" to "zero-mean white noise". Perhaps it is redundant to write that the white noise processes were generated. Authors could consider just saying that white noise was added to the predictor variables.

Response: We agree. Now, it changed to" zero-mean white noises with a mean of 0 and standard deviations of 0.3, 1, and 4 are added to the predictor variables of y2 (or z2) and y4 (or z4).".

Page 9, Lines 171-173. The sentence "The resulting noised series are termed weakly, moderately, 172 and highly noised series respectively, and have a correlation coefficient of 0.9, 0.5, 173 and 0.1 respectively, with their original predictor variable" needs to be rewritten and simplified. Consider breaking the sentence into two separate sentences.

Response: We changed it to two sentences. Now, it looks like "The resulting noised series have correlation coefficients of 0.9, 0.5, and 0.1, respectively, with their original predictor variable. Therefore, we will refer them to weakly, moderately, and highly noised series, respectively." (Lines 177-180 in the attached revision copy).

The authors should carefully check for grammatical errors and make similar changes

throughout the manuscript.

Response: Yes, done. Further English check will be made if a chance for revision will be given.

References Feldstein SB (2000) The timescale, power spectra, and climate noise properties of teleconnection patterns. J Clim 13:4430–4440. doi:10.1175/15200442(2000)0132.0.CO;2 Maraun, D. and Kurths, J.: Cross wavelet analysis: signïficance testingand pitfalls, Nonlin. Processes Geophys., 11, 505–514, 2004. Maraun, D., Kurths, J., and Holschneider, M.: Nonstationary Gaussian processes in wavelet domain: synthesis,estimation, and signïficance testing, Phys. Rev. E, 75,doi:10.1103/PhysRevE.75.016707, 2007. Ng, E. K. W. and Chan, J. C. L.: Geophysical applications of partial wavelet coherence and multiple wavelet coherence, J. Atmos. Ocean. Tech., 29, 1845–1853, doi: 10.1175/JTECH-D-12-00056.1, 2012. Schulte, J. A., Duffy, C., and Najjar, R. G.: Geometric and topological approaches to signïficance testing in wavelet analysis, Nonlin.Processes Geophys., 22, 139–156, doi:10.5194/npg-22-139-2015, 2015. Schulte, J. A.: Cumulative areawise testing in wavelet analysis and its application to geophysical time series, Nonlin. Processes Geophys., 23, 45-57, doi:10.5194/npg-2345-2016, 2016.

Response: Appreciate for the good references. We cited them when we made relevant discussion.

Please also note the supplement to this comment:
http://www.hydrol-earth-syst-sci-discuss.net/hess-2016-154/hess-2016-154-AC2-supplement.pdf

---

## Editor Comment (EC1) · B. Schaefli (Editor) · 28 Jun 2016

The reviewers had only minor comments, to which the authors have responded in this public discussion. I look forward to the revised version, which will be a very interesting technical note for the readership of HESS.

---

## Author Comment (AC5) · 28 Jun 2016

General Comments

The multiple wavelet coherence methodology presented in the manuscript by Hu and Si represents an important contribution to wavelet analysis. In particular, Hu and Si build upon the previous work of Ng and Chan (2012) to extend multiple wavelet coherence to case of more than two predictor variables. The authors further demonstrate that the new multiple wavelet coherence methodology is better suited for situations where the predictor variables are cross-correlated. The problems with the traditional formulation are clearly stated and consistent with the objective of the paper proposed in the introduction section. Theoretical examples were also presented to highlight the advantages of the new methodology relative to existing ones. I their recommend that the manuscript be accepted after the substantial correction of grammatical errors and the consideration of more specific comments presented below.

Response:

Thank you for the positive comments.

Specific comments

The conclusion section simply summarizes the results of the paper. The authors could consider expanding the conclusion section into a discussion section to comment on limitations of the method. After all, wavelet analysis, while useful, is not a scientific panacea. More specifically, the inclusion of more predictor variables may result in the statistical significance threshold at a particular wavelet scale and time to approach unity, which would impose a limit on how much statistical information can be gained. This phenomenon occurs with the traditional multiple wavelet coherence formulation, where the threshold for 5% significance, for example, is higher than that for bivariate wavelet coherence at a given wavelet scale.

Response:

We agree with you that one of the limitation is that the critical values increase with the number of predictor variables. This is also why the percentage area of significant coherence (PASC) for three predictor variables ($z2$, $z4$, and noised $z4$) are even lower than for only two predictor variables ($z2$ and $z4$) when the third predictor variable (noised $z4$) is not statistically significant to explain the variation of the response variable.

We put this limitation in the conclusion part as " Theoretically, any number of predictor variables can be included in the multiple wavelet analysis. However, the statistical significance threshold usually increases with the number of the predictor variables (Grinsted et al., 2004; Ng and Chan, 2012a), and inclusion of too many predictor variables may result in the statistical significance threshold at particular wavelet scales (e.g., the lowest and largest scales) to approach unity. This would restrict the availability of statistical information."

The author may also consider discussing at least briefly the problem of simultaneously testing multiple statistical hypothesis, as discussed in Maraun and Kurths (2004), Maraun et al. (2007), Schulte et al. (2015), and Schulte (2016). Multiple-testing problem is a major problem in wavelet analysis and therefore merits consideration in a discussion section. Presenting clearly the methodological limitations will better guide the likely interdisciplinary readership in making decisions regarding what analysis tools to implement.

Response:

The multiple-testing problem will be briefly discussed in the conclusion part. "In addition, similar to bivariate wavelet analysis, the new method also suffers from the multiple-testing problem (Maraun and Kurths, 2004; Maraun et al., 2007; Schulte et al., 2015; Schulte, 2016). Therefore, a more robust statistical significance testing method may be beneficial to the new method."

Throughout the manuscript, the authors mention how geoscience data are often nonstationary. Perhaps the term is used too loosely in some instances and is sometimes inconsistent with the strict time series analysis definition. Even white and red-noise processes contain time and scale-localized features in wavelet space, even though theirrespectivestatisticsarestationaryatallorders. Time-andscale-localizedfeatures are evident in the wavelet power spectrum of say, the North Atlantic Oscillation (NAO), even though the statistics of the NAO are consistent with a first-order Markov process (Feldstein,2000). Therefore,insomeinstances,Irecommendchangingtheword"nonstationary" to "transient" or "transitory".

Response:

We agree. In the introduction, we will make this more clear as " More often than not, geoscience data are transient, consisting of a variety of frequency regimes that may be localized in space or time (Torrence and Compo, 1998; Si and Zeleke, 2005; Graf et al., 2014). The transient characteristics exists widely in non-stationary but also sometimes in stationary processes (Feldstein, 2000)."

At many instances, we will change the "non-stationary" to "transient" when suitable.

Some Technical Corrections

Page 2 Line 3536. Change "geoscience data is" to "geoscience data are".

Response:

Yes, will change.

Page 2 Line 39. Is it better to say bivariate wavelet coherency rather than "simple wavelet coherency"

Response:

Yes, we will change all throughout the paper.

Page 5, Line 97. Add comma before "respectively".

Response:

Yes, we will change throughout the paper.

Page9,Line169-171. The sentence can be slightly simplified by changing" white noise with a mean of 0" to "zero-mean white noise". Perhaps it is redundant to write that the white noise processes were generated. Authors could consider just saying that white noise was added to the predictor variables.

Response:

We agree. It will be changed to" zero-mean white noises with a mean of 0 and  standard deviations of 0.3, 1, and 4 are added to the predictor variables of $y2$ (or $z2$) and $y4$ (or $z4$).".

Page 9, Lines 171-173. The sentence "The resulting noised series are termed weakly, moderately, 172 and highly noised series respectively, and have a correlation coefficient of 0.9, 0.5, 173 and 0.1 respectively, with their original predictor variable" needs to be rewritten and simplified. Consider breaking the sentence into two separate sentences.

Response:

We will separate it to two sentences. Now, it will look like  "The resulting noised series have correlation coefficients of 0.9, 0.5, and 0.1, respectively, with their original predictor variable. Therefore, we will refer them to weakly, moderately, and highly noised series, respectively."

The authors should carefully check for grammatical errors and make similar changes throughout the manuscript.

Response:

Yes, we will do. English check will be made if a chance for revision will be given.

References

Feldstein SB (2000) The timescale, power spectra, and climate noise properties of teleconnection patterns. J Clim 13:4430–4440. doi:10.1175/15200442(2000)0132.0.CO;2 Maraun, D. and Kurths, J.: Cross wavelet analysis: significance testingand pitfalls, Nonlin. Processes Geophys., 11, 505–514, 2004. Maraun, D., Kurths, J., and Holschneider, M.: Nonstationary Gaussian processes in wavelet domain: synthesis,estimation, and significance testing, Phys. Rev. E, 75,doi:10.1103/PhysRevE.75.016707, 2007. Ng, E. K. W. and Chan, J. C. L.: Geophysical applications of partial wavelet coherence and multiple wavelet coherence, J. Atmos. Ocean. Tech., 29, 1845–1853, doi: 10.1175/JTECH-D-12-00056.1, 2012. Schulte, J. A., Duffy, C., and Najjar, R. G.: Geometric and topological approaches to significance testing in wavelet analysis, Nonlin.Processes Geophys., 22, 139–156, doi:10.5194/npg-22-139-2015, 2015. Schulte, J. A.: Cumulative areawise testing in wavelet analysis and its application to geophysical time series, Nonlin. Processes Geophys., 23, 45-57, doi:10.5194/npg-2345-2016, 2016.

Response:

Appreciate for the good references. We will cite them when we make relevant discussion.

---

## Author Response (AR1)

The manuscript of Multiple wavelet coherence by Hu and Si presented an important topic. In characterizing scale specific variations, wavelet coherence has been used in many field but was restricted to only two variables. Presentation of wavelet coherence produces a step forward on the methodological development aspect. The method will support a lot of different fields including soil science and hydrology. The scientific content is suitable for the journal and the readers of this journal will be interested in this topic. Therefore, my suggestion is for acceptance of the manuscript with some minor corrections such as English, which could be improved. Another thing, authors used the artificial series to compare with other multi-variate analysis. Just wondering, how will you confirm about you claimed superior information of the new method compare to other methods. I mean to say, how will you say that this variations, what is shown by other methods are also showing the right information. The variations showing here could be spurious as identified by different methods.

**Response:**

Thank you for the positive comments.

In terms of language, we have tried our best to correct it, and we have asked an English editing company double check the language.

We are not very sure we understood your second comment, but we will try to explain a bit here. The two existing methods (i.e., multiple spectral coherence and multivariate empirical mode decomposition) are widely used for spatial or temporal series analysis in different disciplines. Actually we have known that these two methods cannot deal with localized relationships between variables. Therefore, the advantages of the new method

over these two methods is demonstrated mainly in terms of relationships between response and predictor variables at various scales of the response variable. The reason for using the artificial data is that the major features (e.g., scale) are known. Then, the superiority of the new method over these two methods can be assessed by whether the known major features of the artificial data are demonstrated by these methods. Our results clearly show that localized multivariate relationships are not available by the two existing methods and both methods are likely to underestimate the degree of multivariate relationships for non-stationary processes. Because the cosine-like artificial datasets mimic many time series and spatial series in geosciences. Therefore, we conclude that the new method is superior.

All above mentioned information can be found in the revised copy. Please refer to them at Lines 82-84, 154-160, 183-187, and 381-384.

**Reply to "Interactive comment on "Technical Note: Multiple wavelet coherence for untangling scale-specific and localized multivariate relationships in geosciences" by W. Hu and B. C. Si " by Referee #2**

**General Comments**

The multiple wavelet coherence methodology presented in the manuscript by Hu and Si represents an important contribution to wavelet analysis. In particular, Hu and Si build upon the previous work of Ng and Chan (2012) to extend multiple wavelet coherence to case of more than two predictor variables. The authors further demonstrate that the new multiple wavelet coherence methodology is better suited for situations where the predictor variables are cross-correlated. The problems with the traditional formulation are clearly stated and consistent with the objective of the paper proposed in the introduction section. Theoretical examples were also presented to highlight the advantages of the new methodology relative to existing ones. I their recommend that the manuscript be accepted after the substantial correction of grammatical errors and the consideration of more specific comments presented below.

**Response:**

**Thank you for the positive comments.**

**Specific comments**

The conclusion section simply summarizes the results of the paper. The authors could consider expanding the conclusion section into a discussion section to comment on limitations of the method. After all, wavelet analysis, while useful, is not a scientific panacea. More specifically, the inclusion of more predictor variables may result in the statistical significance threshold at a particular wavelet scale and time to approach unity,

which would impose a limit on how much statistical information can be gained. This phenomenon occurs with the traditional multiple wavelet coherence formulation, where the threshold for 5% significance, for example, is higher than that for bivariate wavelet coherence at a given wavelet scale.

**Response:**

We agree with you that one of the limitation is that the critical values increase with the number of predictor variables. This is also why the percentage area of significant coherence (PASC) for three predictor variables (z2, z4, and noised z4) are even lower than for only two predictor variables (z2 and z4) when the third predictor variable (noised z4) is not statistically significant to explain the variation of the response variable. Please see Lines 260-261.

We put this limitation in the conclusion part as "Theoretically, any number of predictor variables can be included in the multiple wavelet analysis. However, the statistical significance threshold usually increases with the number of predictor variables (Grinsted et al., 2004; Ng and Chan, 2012a).In addition, the inclusion of too many predictor variables may result in the statistical significance threshold at particular wavelet scales (e.g., the lowest and largest scales) to approach unity. This would restrict the availability of statistical information." (Lines389-395).

The author may also consider discussing at least briefly the problem of simultaneously testing multiple statistical hypothesis, as discussed in Maraun and Kurths (2004), Maraun et al. (2007), Schulte et al. (2015), and Schulte (2016). Multiple-testing problem is a major problem in wavelet analysis and therefore merits consideration in a discussion section. Presenting clearly the methodological limitations will better guide the likely interdisciplinary readership in making decisions regarding what analysis tools to implement.

**Response:**

The multiple-testing problem has been briefly discussed in the conclusion part. "Furthermore, similar to bivariate wavelet analysis, the new method also suffers from the multiple-testing problem (Maraun and Kurths, 2004; Maraun et al., 2007; Schulte et al., 2015; Schulte, 2016). Therefore, a more robust statistical significance testing method may be beneficial to the new method." (Lines395-399).

Throughout the manuscript, the authors mention how geoscience data are often nonstationary. Perhaps the term is used too loosely in some instances and is sometimes inconsistent with the strict time series analysis definition. Even white and red-noise processes contain time and scale-localized features in wavelet space, even though theirrespective statistics are stationary at all orders. Time-and scale-localized features are evident in the wavelet power spectrum of say, the North Atlantic Oscillation (NAO), even though the statistics of the NAO are consistent with a first-order Markov process (Feldstein,2000). Therefore,insomeinstances,Irecommendchangingtheword"nonstationary" to "transient" or "transitory".

**Response:**

We agree. In the introduction, we made this more clear as " More often than not, geoscience data are transient, consisting of a variety of frequency regimes that may be localized in space or time (Torrence and Compo, 1998; Si and Zeleke, 2005; Graf et al., 2014). The transient characteristics exist widely in non-stationary processes, but also sometimes occur in stationary processes (Feldstein, 2000)." (Lines35-39).

At many instances, we changed the "non-stationary" to "transient" when suitable, such as Line 42, 58, 65 in the attached revision.

Some Technical Corrections

Page 2 Line 3536. Change "geoscience data is" to "geoscience data are".

Response:

Yes, done at L36.

Page 2 Line 39. Is it better to say bivariate wavelet coherency rather than "simple wavelet coherency"

**Response:**

Yes, we changed all throughout the paper.

Page 5, Line 97. Add comma before "respectively".

Response:

Yes, we did throughout the paper.

Page9,Line169-171. The sentence can be slightly simplified by changing" white noise with a mean of 0" to "zero-mean white noise". Perhaps it is redundant to write that the white noise processes were generated. Authors could consider just saying that white noise was added to the predictor variables.

**Response:**

We agree. Now, it changed to "zero-mean white noises with standard deviations of 0.3, 1, and 4 are added to the predictor variables of y2 (or z2) and y4 (or z4)".

Page 9, Lines 171-173. The sentence "The resulting noised series are termed weakly, moderately, 172 and highly noised series respectively, and have a correlation coefficient of 0.9, 0.5, 173 and 0.1 respectively, with their original predictor variable" needs to be rewritten and simplified. Consider breaking the sentence into two separate sentences.

**Response:**

We changed it to two sentences. Now, it is like "The resulting noised series have correlation coefficients of 0.9, 0.5, and 0.1, respectively, with their original predictor variable. Therefore, we will refer to them as weakly, moderately, and highly noised series, respectively." (Lines 175-177).

The authors should carefully check for grammatical errors and make similar changes throughout the manuscript.

**Response:**

**Yes, done.**

We have asked an English editing company double check the language.

**Response:**

Appreciate for the good references. We cited them when we made relevant discussion.

[revised manuscript text omitted]

102 can be written as

103
$$\overline{W}^{X,X}(s,\tau) = \begin{bmatrix} \overline{W}^{X_{1,X_{1}}}(s,\tau) & \overline{W}^{X_{1,X_{2}}}(s,\tau) & \cdots & \overline{W}^{X_{1,X_{q}}}(s,\tau) \\ \overline{W}^{X_{2,X_{1}}}(s,\tau) & \overline{W}^{X_{2,X_{2}}}(s,\tau) & \cdots & \overline{W}^{X_{2,X_{q}}}(s,\tau) \\ \vdots & \vdots & & \vdots \\ \overline{W}^{X_{q,X_{1}}}(s,\tau) & \overline{W}^{X_{q,X_{2}}}(s,\tau) & \cdots & \overline{W}^{X_{q,X_{q}}}(s,\tau) \end{bmatrix},$$
(1)

104 where— $\overrightarrow{W}^{X_i,X_j}(s,\tau)$  is the smoothed auto-wavelet power spectra (when i=j) or 105 cross-wavelet power spectra (when  $i\neq j$ ) at scale *s* and spatial (or temporal) location Formatted: Font: (Default) Times New Roman, 12 pt, Not Bold, Font color: Auto

 $\tau_{\star}$  respectively. For the detailed calculation of smoothed auto- and cross-wavelet

107 power spectra, see Supplement, Sect. S1.

108 The matrix of smoothed cross wavelet power spectra between response variable Y

and predictor variables Xi can be defined as

110
$$\overrightarrow{W}^{Y,X}(s,\tau) = \left[ \overrightarrow{W}^{Y,X1}(s,\tau) \quad \overrightarrow{W}^{Y,X2}(s,\tau) \quad \cdots \quad \overrightarrow{W}^{Y,Xq}(s,\tau) \right],$$
(2)

111 where  $\overrightarrow{W}^{Y,Xi}(s,\tau)$  is the smoothed cross-wavelet power spectra between Y and Xi at 112 scale s and spatial (or temporal) location  $\tau$ .

113 The smoothed wavelet power spectrum of response variable Y is  $\overline{W}^{Y,Y}(s,\tau)$ . 114 Following Koopmans (1974), the MWC at scale s and location  $\tau$ ,  $\rho_m^{(2)}(s,\tau)$ , can

115 be written as

116
$$\rho_{m}^{2}(s,\tau) = \frac{\overline{W}^{Y,X}(s,\tau)\overline{W}^{X,X}(s,\tau)^{-1}\overline{\overline{W}^{Y,X}(s,\tau)}}{\overline{W}^{Y,Y}(s,\tau)}.$$
(3)

117 When only one predictor variable (e.g., *XI*) is included in *X*, Eq. (3) is the equation 118 for bivariatesimple wavelet coherence,  $\rho_b^2(s,\tau) = \rho_s^2(s,\tau)$ , between two 119 variableswhich can be expressed as (Torrence and Webster, 1999; Grinsted et al., 120 2004):

121

$$\rho_b^2(s,\tau) = \frac{\overrightarrow{W}^{Y,X1}(s,\tau)\overrightarrow{W}^{Y,X1}(s,\tau)}{\overrightarrow{W}^{X1,X1}(s,\tau)\overrightarrow{W}^{Y,Y}(s,\tau)}.$$
(4)

Therefore, bivariatesimple wavelet coherence is consistent with multiple wavelet coherence if only one predictor variable is included. In addition, the wavelet phase between a response variable (Y) and a predictor variable (XI) is **Field Code Changed**

**Field Code Changed**

**Field Code Changed**

125

[revised manuscript text omitted]
\_For-the stationary case\_for example, the MCCmemd values at the scales corresponding to  $y_2$  (or and  $y_4$ ) decrease from 0.98 to 0.49 and from 0.93 to 0.62 when the second half of the  $y_2$  (or  $y_4$ ) series are replaced by 0. , and from 0.93 to 0.62, respectively, when the second half of the  $y_2$  and  $y_4$  series are replaced by 0.

349 As explained above, the MWC has advantages in untangling localized multivariate relationships as compared to the common multivariate methods. It is important to 350 reveal the multivariate relationships; which vary with time or space, that are 351 352 associated with different processes. For example, discharge usually happens occurs on knolls, while recharge usually happens occurs in neighboring depressions (Gates et al., 353 354 2011). Therefore, the controlling factors of soil water storage may vary with the land 355 element characteristics of a location. For example, ILocal controls may be more 356 important on knolls, while non-local controls may be more important in depressions 357 (Grayson et al., 1997). In a temporal domain, vegetation transpiration contributes more to the evapotranspiration in the growing seasons, which may result in the 358 changes of environmental factors explaining temporal variations of evapotranspiration 359 in different seasons. 360

**361 **4.4 Application of the MWC**

Each meteorological factor was significantly correlated to the *E*, but the dominant factors explaining variations in *E* differed with scale. For example, the relative humidity was the dominating factor at small (2–8 months) and large (>32 months) scales, while temperature was the dominating factor at the medium (8–32 months) scales. Overall, the relative humidity corresponded to the greatest mean MWC (0.62) and PASC value (40%) at multiple scale-location domains. For the detailed relationships between *E* and each factor, see Supplement, Sect. S6.

The MWC analysis shows that the combination of relative humidity and mean 369 temperature produced the greatest mean MWC (0.82) and PASC (49%) among all 370 two-factor cases.5 This indicatsuggesteding that relative humidity and mean 371 temperature they are were the best most appropriate factors to for explaining variations 372 373 in E at multiple scale-location domains (Fig. 7a). However, adding an additional 374 factor such as sun hours, which was the best among all three-factor cases, increased 375 the average coherence (0.91), but slightly decreased the PASC to 48% (Fig. 7b). This 376 indicated that sun hours was not significantly different from red noise in explaining additional variation in E. Similar results were found when the wind speed was added. 377 378 The This occurs because reason for this was being that most areas with significant coherence between E and sun hours or wind speed, were a subset of areas with 379 380 significant coherence between E and relative humidity or mean temperature (see Supplement, Sect. S3). Therefore, relative humidity and mean temperature were 381 adequate  $\frac{1}{10}$  explaining the temporal variation of E at various scales at this site. 382

This is-was consistent with Li et al. (2012), who indicated that relative humidity and 383 mean temperature are were the two main contributors to the temporal change of 384 385 potential evapotranspiration on the Chinese Loess Plateau.

**5. Conclusions 386**

387 Multiple wavelet coherence is-was\_developed to determine scale-specific and 388 localized multivariate relationships in geosciences. The new method is-was tested and 389 compared with existing multivariate methods, using an artificial dataset. The new 390 method can be used to determine the proportion of the variance of a response variable 391 that is explained by predictor variables, at a specific scale and location (spatially or temporally). As compared with bivariatesimple wavelet coherence, more variation 392 may be explained at multiple scale-location domains by the MWC. Including more 393 variables is only beneficial if the variables are not strongly cross-correlated, and can 394 395 independently explain a fair amount of variability in a response variable. Therefore, the best combinations of variables that explain multivariate, spatial or temporal 396 variability at multiple scales can be determined. This is important for optimizing 397 398 variables for to developing scale-specific prediction.

The MSC and MCCmemd can determine multivariate relationships at multiple scales, 399 but localized multivariate relationships are not available. Furthermore, and-both MSC 400 401 and MCCmemd are likely to underestimate the degree of multivariate relationships for 402 non-stationary processes. In addition, the performance of MCCmend relies on the 403 performance of MEMD, which needs further development. Application of the MWC

| 404 | into the real dataset indicates that the combination of relative humidity and mean         |
|-----|--------------------------------------------------------------------------------------------|
| 405 | temperature are the optimal factors that can be used to explain temporal variations of     |
| 406 | E at the Changwu site in China.                                                            |
| 407 | Limitations of the new method also exist. Theoretically, any number of predictor           |
| 408 | variables can be included in the multiple wavelet analysis. However, the statistical       |
| 409 | significance threshold usually increases with the number of the predictor variables        |
| 410 | (Grinsted et al., 2004; Ng and Chan, 2012a). , and iIn addition, the inclusion of too      |
| 411 | many predictor variables may result in the statistical fisignetic threshold at             |
| 412 | particular wavelet scales (e.g., the lowest and largest scales) to approach unity. This    |
| 413 | would restrict the availability of statistical information. In additionFurthermore,        |
| 414 | similar to bivariate wavelet analysis, the new method also suffers from the                |
| 415 | multiple-testing problem (Maraun and Kurths, 2004; Maraun et al., 2007; Schulte et         |
| 416 | al., 2015; Schulte, 2016). Therefore, a more robust statistical significance testing       |
| 417 | method may be beneficial to the new method.                                                |
| 418 | In summary, multiple wavelet coherence has advantages over existing multivariate           |
| 419 | methods, and provides an effective vehicle for untangling complex spatial or temporal      |
| 420 | variability for multiple controlling factors at multiple scales and locations. It may also |
| 421 | be used as a data-driven tool for modeling and predicting various processes in the area    |
| 422 | of geosciences, such as precipitation, drought, soil water dynamics, stream flow, and      |
| 423 | atmospheric circulation.                                                                   |

**424 Acknowledgements**

| 425 | The Matlab        | codes for cal   | culating multiple w  | avelet cohere | ence are develo     | ped base   | d on  |
|-----|-------------------|-----------------|----------------------|---------------|---------------------|------------|-------|
| 426 | the               | codes           | provided             | by            | А.                  | Grir       | isted |
| 427 | (http://noc.      | ac.uk/using-sc  | ience/crosswavelet-  | wavelet-cohe  | erence) and, t      | ogether    | with  |
| 428 | the user m | anual, are av   | ailable in the Supp  | lement (Sec   | t. S2-S4). The      | e project  | was   |
| 429 | partially fu      | nded by the Na  | atural Sciences and  | Engineering   | Research Coun       | icil of Ca | nada  |
| 430 | (NSERC) a         | and Agricultur  | e Development Fu     | nd of Saska   | chewan. We t | hank the   | two   |
| 431 | anonymous         | s reviewers for | their constructive c | omments.      |                     |            |       |

[revised manuscript text omitted]

---

## Author Response (AR2)

Dear Prof. Schaefli,

Thank you very much for your comments again.

We have made the following corrections for this corrected copy:

(1) We shortened the figure captions for not repeating information.

However, we still repeat the explanations of some variables to make each figure relatively independent.

(2) Thank you for suggesting us a good paper. We cited it at Line 396 when discuss the multiple testing problem.

(3) We have made all variable names in italic. These changes have covered the whole manuscript including text, equations, figures, figure captions, tables, and supplement.

(4) We have figured out the latex problem at Line 392 (previous Line 393).

(5) In the S4 of supplement, we provide the names of all Matlab codes included in the package provided by A. Grinsted which can be available from http://www.glaciology.net/wavelet-coherence. Please see at Lines 588-590 which reads as " In the package provided by A. Grinsted, Matlab codes included are *anglemean.m*, *ar1.m*, *ar1nv.m*, *boxpdf.m*, *formatts.m*, *normalizepdf.m*, *phaseplot.m*, *smoothwavelet.m*, *wt.m*, *wtc.m*, *wtcdemo.m*, *wtcsignif.m*, *xwt.m*. ". By this way, readers can find what they need if the website will be updated.

(6) In the acknowledgement part, we added two funding supporters. So "The project was partially funded by the Natural Sciences and Engineering Research Council of Canada (NSERC) and Agriculture Development Fund of Saskatchewan." was changed to " The project was funded by the National Natural Science Foundation of China (41371233), the Natural Sciences and Engineering Research Council of Canada (NSERC), Agriculture Development Fund of Saskatchewan, and the New Zealand Institute for Plant & Food Research under the Land Use Change and Intensification programme"

(7) In the S2-S4 of the supplement, we suggest readers cite this work if they used our codes for publication. We leave the full citation of this manuscript and website of the supplement in red to be determined at Lines 101, 104, 394, 397, 596, and 602. Would you please be able to figure this out for me before publication? Or would you please guide me how to figure it out by ourselves?

The marked-up manuscript is attached in the following.

We hope this manuscript have reached the level for publication.

Thank you so much again.

Sincerely,

Wei Hu and Bing Si

[revised manuscript text omitted]

98
$$\overline{W}^{X,X}(s,\tau) = \begin{bmatrix} \overline{W}^{X_1,X_1}(s,\tau) & \overline{W}^{X_1,X_2}(s,\tau) & \cdots & \overline{W}^{X_1,X_q}(s,\tau) \\ \overline{W}^{X_2,X_1}(s,\tau) & \overline{W}^{X_2,X_2}(s,\tau) & \cdots & \overline{W}^{X_2,X_q}(s,\tau) \\ \vdots & \vdots & & \vdots \\ \overline{W}^{X_q,X_1}(s,\tau) & \overline{W}^{X_q,X_2}(s,\tau) & \cdots & \overline{W}^{X_q,X_q}(s,\tau) \end{bmatrix},$$
 (1)

99 where  $\overrightarrow{W}^{x_i, x_j}(s, \tau)$  is the smoothed auto-wavelet power spectra (when i=j) or 100 cross-wavelet power spectra (when  $i\neq j$ ) at scale *s* and spatial (or temporal) location 101  $\tau$ , respectively. For the detailed calculation of smoothed auto- and cross-wavelet 102 power spectra, see Supplement, Sect. S1.

103 The matrix of smoothed cross wavelet power spectra between response variable Y

- 104
- and predictor variables  $X_i$  can be defined as

105
$$\overrightarrow{W}^{Y,X}(s,\tau) = \left[\overrightarrow{W}^{Y,X_1}(s,\tau) \quad \overrightarrow{W}^{Y,X_2}(s,\tau) \quad \cdots \quad \overrightarrow{W}^{Y,X_q}(s,\tau) \right],$$
(2)

5

106 where  $\overline{W}^{Y,X_i}(s,\tau)$  is the smoothed cross-wavelet power spectra between Y and  $X_i$  at 107 scale s and spatial (or temporal) location  $\tau$ . 108 The smoothed wavelet power spectrum of response variable Y is  $\overline{W}^{Y,Y}(s,\tau)$ . 109 Following Koopmans (1974), the MWC at scale s and location  $\tau$ ,  $\rho_m^{-2}(s,\tau)$ , can

110 be written as

111
$$\rho_{m}^{2}(s,\tau) = \frac{\overline{W}^{Y,X}(s,\tau)\overline{W}^{X,X}(s,\tau)^{-1}\overline{W}^{Y,X}(s,\tau)}{\overline{W}^{Y,Y}(s,\tau)}.$$
(3)

112 When only one predictor variable (e.g.,  $X_{a}$ ) is included in *X*, Eq. (3) is the equation 113 for bivariate wavelet coherence,  $\rho_b^2(s,\tau)$ , which can be expressed as (Torrence and 114 Webster, 1999; Grinsted et al., 2004):

115
$$\rho_b^2(s,\tau) = \frac{\overline{W}^{Y,X_1}(s,\tau)}{\overline{W}^{X_1,X_1}(s,\tau)\overline{W}^{Y,Y}(s,\tau)}.$$
(4)

116 Therefore, bivariate wavelet coherence is consistent with multiple wavelet 117 coherence if only one predictor variable is included. In addition, the wavelet phase 118 between a response variable (Y) and a predictor variable ( $X_{1}$ ) is

119
$$\phi(s,\tau) = \tan^{-1}\left(\operatorname{Im}\left(W^{Y,X_{1}}(s,\tau)\right) / \operatorname{Re}\left(W^{Y,X_{1}}(s,\tau)\right)\right), \qquad (5)$$

where Im and Re denote the imaginary and real part of  $W^{Y,X_1}(s,\tau)$ , respectively. Note that the phase information between a response variable *Y* and multiple predictor variables *X* cannot be obtained.

Multiple wavelet coherence at the 95% confidence level is calculated using the Monte Carlo method (Grinsted et al., 2004). Surrogate spatial series (i.e., red noise) of all variables are generated with a Monte Carlo simulation based on their first-order autocorrelation coefficient (AR1). The MWC at each scale and location is calculated Formatted: Subscript

using the simulated spatial series. This is repeated an adequate number of times (e.g.,
1000) (Grinsted et al., 2004). At each scale, MWCs at all locations outside the cones
of influence, from all simulations are ranked in ascending order. The value at the 95th
percentile represents the 95% confidence level for the MWC at that scale. The Matlab
codes and user manual document for calculating MWC and significance level are
provided in the Supplement (Sect. S2–S4).

**133 **3. Data and analysis**

**134 **3.1 Artificial data for method test**

| 135 | The method is tested using a stationary and non-stationary artificial dataset,                                        |
|-----|-----------------------------------------------------------------------------------------------------------------------|
| 136 | generated following Yan and Gao (2007). The response variable (y for the stationary                                   |
| 137 | case and z for the non-stationary case) encompasses five cosine waves ( $y_1$ to $y_5$ for the                        |
| 138 | stationary case and $z_{1}$ to $z_{5}$ for the non-stationary case), with different dimensionless                     |
| 139 | scales (Fig. 1). For the stationary case, $y_1 = \cos(2\pi x/4)$ , $y_2 = \cos(2\pi x/8)$ , $y_3 = \cos(2\pi x/16)$ , |
| 140 | $y_4 = \cos(2\pi x/32)$ , and $y_5 = \cos(2\pi x/64)$ , where $x=0, 1, 2,, 255$ . There is one regular                |
| 141 | cycle every 4, 8, 16, 32, and 64 locations, representing dimensionless scales of 4, 8,                                |
| 142 | 16, 32, and 64 for $y_1$ , $y_2$ , $y_3$ , $y_4$ , and $y_5$ , respectively (Fig. 1a). The regular cycles             |
| 143 | make each predictor and response series stationary. For the non-stationary case,                                      |
| 144 | $z_1 = \cos(500\pi(x/1000)^{0.5}), \qquad z_2 = \cos(250\pi(x/1000)^{0.5}), \qquad z_3 = \cos(125\pi(x/1000)^{0.5}),$ |
| 145 | $z_4 = \cos(62.5\pi(x/1000)^{0.5})$ , and $z_5 = \cos(31.25\pi(x/1000)^{0.5})$ , where x=0, 1, 2,, 255.               |
| 146 | The equation containing the square root of the location term results in the gradual                                   |
| 147 | change in frequency (scale), with the greatest dimensionless scales of 4, 8, 16, 32, and                              |

Formatted

| 148 | 64 at the right hand side for $z_1$ , $z_2$ , $z_3$ , $z_4$ , and $z_5$ , respectively (Fig. 1b). The average |
|-----|---------------------------------------------------------------------------------------------------------------|
| 149 | scales for these predictor variables are 3, 5, 9, 17, and 32, respectively. The                               |
| 150 | location-varying scales make each predictor and response variable non-stationary.                             |
| 151 | For both the stationary and non-stationary series, the variance of the response                               |
| 152 | variable is 2.5. The predictor variables, each with a variance of 0.5, are orthogonal to                      |
| 153 | each other, and contribute equally to the total variance of the response variable. The                        |
| 154 | cosine-like artificial datasets mimic many time series such as seismic signals,                               |
| 155 | turbulence, air temperature, precipitation, hydrologic fluxes, and the El                                     |
| 156 | Niño-Southern Oscillation. They also mimic geoscientific spatial series such as ocean                         |
| 157 | waves, seafloor bathymetry, land surface topography, and soil water content along a                           |
| 158 | hummocky landscape. Therefore, they are representative of a geoscience data series                            |
| 159 | and are suitable for testing the new method.                                                                  |
| 160 | Multiple wavelet coherence between the response variable $y$ (or $z$ ) and two ( $y_2$ and                    |

 $y_4$ , or  $z_2$  and  $z_4$ ) or three  $(y_2, y_3, y_4, y_4, y_4, y_5, z_3, y_4, y_4)$  predictor variables were 161 calculated. The advantage of the artificial data is that the known scale- and localized 162 163 features for all variables, and the known relationships between the response and each 164 predictor variable, are exact. By definition, the coherence is 1 at scales corresponding to those of the included predictor variables, and 0 at other scales. 165

To demonstrate the advantages of MWC in dealing with abrupt changes (a type of 166 transient and localized feature), the second half of the original series of  $y_2$  (or  $z_2$ ) or  $y_4$ 167 (or  $z_4$ ) are replaced by 0, and MWC between the response variable and new set of 168 predictor variables is calculated. We anticipate that the coherence changes from 1 to 0 169

| Formatted: Font: Italic |
|-------------------------|
| Formatted: Subscript    |
| Formatted: Font: Italic |
| Formatted: Subscript    |
| Formatted: Font: Italic |
| Formatted: Subscript    |
| Formatted: Font: Italic |
| Formatted: Subscript    |
| Formatted: Font: Italic |
| Formatted: Subscript    |

| I      | Formatted: Font: Italic |
|--------|-------------------------|
| ļ      | Formatted: Font: Italic |
| ļ      | Formatted: Font: Italic |
| 1      | Formatted: Subscript    |
| 1      | Formatted: Font: Italic |
| 1      | Formatted: Subscript    |
| 1      | Formatted: Font: Italic |
| /      | Formatted: Subscript    |
| /      | Formatted: Font: Italic |
| /      | Formatted: Subscript    |
| /      | Formatted: Font: Italic |
| /      | Formatted: Subscript    |
|        | Formatted: Font: Italic |
| _      | Formatted: Subscript    |
|        | Formatted: Font: Italic |
|        | Formatted: Subscript    |
| \
\ | Formatted: Font: Italic |
| ۱
۱ | Formatted: Subscript    |
| ١      | Formatted: Font: Italic |
| ۱      | Formatted: Subscript    |
| ľ      | Formatted: Font: Italic |
| 1      | Formatted: Subscript    |
| /      | Formatted: Font: Italic |
| /      | Formatted: Subscript    |
| -      | Formatted: Font: Italic |
|        | Formatted: Subscript    |
|        | Formatted: Font: Italic |
| ١      | Formatted: Subscript    |
|        | Formatted: Font: Italic |
| ١      | Formatted: Subscript    |
|        |                         |

at the location where the new predictor variable becomes 0.

171 Predictor variables may not be as regular as that shown in Fig. 1, and may also be 172 cross-correlated to one another. For these reasons, zero-mean white noises with standard deviations of 0.3, 1, and 4 are added to the predictor variables of  $y_2$  (or  $z_2$ ) 173 and  $y_4$  (or  $z_4$ ). The resulting noised series have correlation coefficients of 0.9, 0.5, and 174 0.1, respectively, with their original predictor variable. Therefore, we will refer to 175 them as weakly, moderately, and highly noised series, respectively. Multiple wavelet 176 177 coherences between the response variable and different predictor variables (original and noised series) are calculated to demonstrate the performance of MWC when 178 noised or correlated predictor variables are involved. Only the non-stationary case 179 will be demonstrated, because the performances of MWC for stationary and 180 non-stationary cases are similar. 181

The MWC is compared to the MSC (Koopmans, 1974; Si, 2008) and MCCmemd 182 183 (Hu and Si, 2013), which are widely used for spatial or temporal series analysis in 184 different disciplines. The advantages of the new method over these two methods will 185 be demonstrated mainly in terms of relationships between response and predictor variables at various scales of the response variable. The MSC is calculated based on 186 the calculated auto- and cross- power spectra, using an equation similar to Eq. (3). 187 188 The detailed introduction of this method can be found in Si (2008). For the calculation of MCCmend, a set of response and predictor variables form a multivariate data series 189 for MEMD. The MEMD is a data driven method and has the ability to align "common 190 scales" present within multivariate data. Please refer to Rehman and Mandic (2010) 191

| Formatted: Font: Italic |
|-------------------------|
| Formatted: Subscript    |
| Formatted: Font: Italic |
| Formatted: Subscript    |
| Formatted: Font: Italic |
| Formatted: Subscript    |
| Formatted: Font: Italic |
| Formatted: Subscript    |

Hu and Si (2013) for the MEMD analysis, 192 and and the website (http://www.commsp.ee.ic.ac.uk/~mandic/research/emd.htm) for the related Matlab 193 194 codes. The original series of response and predictor variables can be decomposed by the MEMD, into different components (IMFs) with varying scales. For IMFs at the 195 196 same scale, multiple stepwise regressions are conducted between response and predictor variables, and the multiple correlation coefficients for each scale-specific 197 IMF are calculated. 198

**199 **3.2 Real data for application**

Daily evaporation (*E*) from free water surfaces in an E601 evaporation pan (pan diameter of 61.8 cm), and other meteorological factors (i.e., relative humidity, mean temperature, sun hours, and wind speed) were collected from January 1, 1979 to December 31, 2013, at Changwu site in Shaanxi, China. The Changwu site is a transition area between semi-arid and subhumid climates, where agricultural productivity is mainly limited by water. Monthly averages of all variables were used in this study, because we are mainly interested in seasonal and inter-annual variability.

**207 4. Results and discussion**

**208 4.1 MWC with orthogonal predictor variables**

For the stationary data, there are two narrow, horizontal bands (red color) representing an MWC value of around 1, at the respective scales of 8 and 32 for all locations (Fig. 2a). These two bands also correspond to the scales of 8 and 32, respectively, for the two predictor variables. When an additional predictor variable with the scale of 16 is introduced, a wide band appears from 6 to 40, signifying that the MWC equals approximately 1 at all locations, at the scales of 8, 16, and 32. As anticipated, when all five predictor variables with scales ranging from 4 to 64 are included, coherence values of close to 1 are found in the whole scale-location domain (data not shown).

The application of MWC to the non-stationary datasets shows that the scales with 218 significant MWC values gradually increase as distance increases. This increase in the 219 scales is due to the non-stationarity of the variables (Fig. 2b). For example, when 220 221 predictor variables of  $z_2$  and  $z_4$  are included, scales of the two bands corresponding to MWC around 1 increase from 4 to 8 and from 8 to 32, respectively. Furthermore, as 222 expected, for only one predictor variable (stationary and non-stationary), MWC 223 reduces to bivariate wavelet coherence; there is only one band of coherence around 1, 224 225 which corresponds to the scale of that predictor variable (data not shown). Note that the significant MWC values for both stationary and non-stationary cases are not 226 227 exactly 1 at all scales or locations, due to the smoothing effect along both scales and locations. However, the mean MWC values of the significant bands are very high (i.e., 228 0.94–1.00), and the MWC values at the centre of the significant band are 1, which 229 230 corresponds to the exact scale of a predictor variable.

When the point values in the second half of the data series of a predictor variable are replaced by 0, the MWC values in that half of the data series are almost 0 at scales corresponding to that predictor variable (Fig. 3). For the stationary case, when the Formatted: Font: Italic Formatted: Subscript Formatted: Font: Italic Formatted: Subscript point values in the second half of the data series of predictor variable  $\underline{y_2}$  (or  $\underline{y_4}$ ) are replaced by 0, the MWC values are around 1 at the scale of 8 (or 32) in the first half of the transect, and 0 in the second half (Fig. 3a). Similar results are also found for the non-stationary case (Fig. 3b). This is expected because the constant series of 0 is not correlated to the response variables at any scale. Much like bivariate wavelet coherence, the MWC method is able to detect abrupt changes in the data series, and has the advantages of dealing with localized multivariate relationships.

241 4.2 MWC with noised and correlated predictor variables

| 242 | When $z_2$ and a noised series derived from $z_2$ are included as predictor variables,           |
|-----|--------------------------------------------------------------------------------------------------|
| 243 | there is only one band of coherence close to 1 at scales corresponding to $z_2$ ,                |
| 244 | irrespective of the correlation between $z_2$ and a noised series of $z_2$ (Fig. 4a). When $z_2$ |
| 245 | and a noised series of $z_4$ are included as predictor variables, the coherence depends on       |
| 246 | the degree of the noise (Fig. 4b). For weakly noised series, there are two bands of              |
| 247 | coherence of around 1, corresponding to the scales of $z_2$ and $z_4$ , respectively. The        |
| 248 | percentage area of significant coherence (PASC) is 23%, which equals that of when $z_2$          |
| 249 | and $z_{4}$ are included. With the increasing magnitude of noise, the coherence and              |
| 250 | corresponding PASC at the scales corresponding to $z_4$ decrease. When $z_2$ and a               |
| 251 | strongly noised series of $z_4$ are considered, the band of coherence around 1, at scales        |
| 252 | corresponding to $Z_4$ , disappears.                                                             |
| 253 | The inclusion of a third noised $z_4$ variable substantially increases the area with high        |
| 254 | coherence (in red) as compared to the case when only $z_2$ and $z_4$ are included (Fig. 4c).     |

| Formatted: Font: Italic |
|-------------------------|
| Formatted: Subscript    |
| Formatted: Font: Italic |
| Formatted: Subscript    |

| Formatted: Font: Italic |  |
|-------------------------|--|
| Formatted: Subscript    |  |
| Formatted: Font: Italic |  |
| Formatted: Subscript    |  |
| Formatted: Font: Italic |  |
| Formatted: Subscript    |  |
| Formatted: Font: Italic |  |
| Formatted: Subscript    |  |
| Formatted: Font: Italic |  |
| Formatted: Subscript    |  |
| Formatted: Font: Italic |  |
| Formatted: Subscript    |  |
| Formatted: Font: Italic |  |
| Formatted: Subscript    |  |
| Formatted: Font: Italic |  |
| Formatted: Subscript    |  |
| Formatted: Font: Italic |  |
| Formatted: Subscript    |  |
| Formatted: Font: Italic |  |
| Formatted: Subscript    |  |
| Formatted: Font: Italic |  |
| Formatted: Subscript    |  |
| Formatted: Font: Italic |  |
| Formatted: Subscript    |  |
| Formatted: Font: Italic |  |
| Formatted: Subscript    |  |
| Formatted: Font: Italic |  |
| Formatted: Subscript    |  |
| Formatted: Font: Italic |  |
| Formatted: Subscript    |  |
| Formatted: Font: Italic |  |
| Formatted: Subscript    |  |
| Formatted: Font: Italic |  |
| Formatted: Subscript    |  |
| Formatted: Font: Italic |  |
| Formatted: Subscript    |  |

This indicates that MWC will increase as the number of predictor variables increases, 255 with the highest coherence less or equal to 1, irrespective of the number of predictor 256 257 variables. However, the area of significant coherence may not necessarily increase because of the simultaneously increased statistical significance threshold (Ng and 258 259 Chan, 2012a). In fact, the PASC values for three predictor variables (19–20%) are lower than those of the two predictor variables (23%). This indicates that, in this case, 260 261 two predictor variables are better than three in terms of explaining the variations of 262 the response variable. This occurs because the variance of the response variable that is explained by the noised variable is already accounted for by other variables. Therefore, 263 264 only an additional variable that can independently explain a fair amount of variance could contribute significantly to explaining variations of a response variable (Fig. 4b). 265 This may also explain why there is only one band of coherence around 1 at scales 266 corresponding to  $z_2$ , when  $z_2$  and a noised series of  $z_2$  are included (Fig. 4a). This 267 268 information is helpful in choosing predictor variables for developing scale-specific predictions, especially when predictor variables are correlated. 269

**270 **4.3 Comparison with other multivariate methods**

271 4.3.1 MSC

The MSC as a function of scale is shown in Fig. 5a. For the stationary case, when  $y_2$  and  $y_4$  are included as predictor variables, there are two plateaus centered at the scales of 8 and 28, representing a coherence of 1. As expected, when an additional predictor variable  $y_3$  is added, the corresponding scale of 16 also shows coherence of

| 1 | Formatted: Font: Italic |
|---|-------------------------|
| - | Formatted: Subscript    |
| Ľ | Formatted: Font: Italic |
|   | Formatted: Subscript    |
|   | Formatted: Font: Italic |
|   | Formatted: Subscript    |
|   |                         |

| 1  | Formatted: Font: Italic |
|----|-------------------------|
| -{ | Formatted: Subscript    |
| ٦  | Formatted: Font: Italic |
| ١  | Formatted: Subscript    |
| 1  | Formatted: Font: Italic |
| -{ | Formatted: Subscript    |

1. The MSC produces similar scale-specific relationships, as MWC does for a 276 stationary dataset, with exception given to the centered scale (i.e., 28) with a 277 278 coherence of 1. Here, the scale with a unity MSC deviates from the expected value 279 (i.e., 32) for predictor variable  $y_4$ . For the non-stationary case, however, the MSC is 280 much lower than 1 for the predictor variables of  $z_2$  and  $z_4$ ; an MSC of 1 is present only at the scale of 8 when an additional predictor variable z3 is added. Obviously, the 281 MSC underestimates the multivariate relationships, and is not suitable for 282 non-stationary processes (Si, 2008) due to its inability to deal with localized features. 283 The MSC at a specific scale provides the average of multivariate relationships, across 284 285 all locations. Due to the change in scale of a predictor variable with location for the non-stationary case, the MSC deviates greatly from 1. 286

The MSC decreases at scales when the second half of the included predictor 287 variable series are replaced by 0 for both the stationary and non-stationary series (Fig. 288 289 5b). For example, when the second half of the  $y_4$  series in the stationary case are replaced by 0, the MSC at scales of around 32 decreases from 1 to 0.52. Although the 290 291 MSC, throughout the second half of the series, can detect the decrease of coherence at the scales corresponding to the 0 values, the exact locations for the decrease cannot be 292 293 identified. In fact, the coherence decreases only in the second half of the series, and 294 does not change in the first half of the series. The location for the decrease can be easily identified by the MWC, but not by MSC. This further demonstrates the 295 inability of the MSC to deal with localized features. 296

| Formatted: Font: Italic |
|-------------------------|
| Formatted: Subscript    |
| Formatted: Font: Italic |
| Formatted: Subscript    |
| Formatted: Font: Italic |
| Formatted: Subscript    |
| Formatted: Font: Italic |
| Formatted: Subscript    |
|                         |

**297 4.3.2 MCCmemd**

298 Five intrinsic mode functions (IMFs) with non-negligible variance, are obtained for multivariate data series. While the obtained scales for the response variable y are in 299 300 agreement with the true scales for the stationary case, the obtained scales (i.e., 3, 6, 11, 21, and 43) for the response variable z deviate slightly from the average scales for the 301 non-stationary case. For the response variable, the contribution of IMFs to the total 302 variance generally decreases (20% to 13% for stationary, and 27% to 11% for 303 304 non-stationary) from IMF1 to IMF5. This disagrees with the fact that each scale contributes equally (i.e., 20%) to the total variance. In addition, the sum of variances 305 over all IMFs for each variable is less than 100% (ranging from 84% to 93%), 306 307 indicating that MEMD cannot capture all the variances. For the detailed results of MEMD, see Supplement, Sect. S5. 308

309 The MCCmemd as a function of scale, is shown in Fig. 6a. For the stationary case, 310 when predictor variables of  $y_2$  and  $y_4$  are included, the MCCmemd values are 0.98 and 311 0.93, respectively, at scales corresponding to those of  $y_2$  and  $y_4$ . When a predictor 312 variable of y3 is included, the MCCmemd values are 1.00, 1.00, and 0.96, respectively, at scales corresponding to those of  $y_2, y_3$ , and  $y_4$ . For the non-stationary, two predictor 313 variable case, the corresponding MCCmend values are 0.80 and 0.85. For the 314 315 non-stationary, three predictor variable case, the corresponding MCCmend values are 0.95, 0.99, and 0.91, respectively. Therefore, the MCCmemd can be used to determine 316 317 the scale-specific multivariate relationships. Similar to MSC, however, the MCCmend underestimates the multivariate relationships, especially for the non-stationary case 318

| Formatted: Font: Italic |
|-------------------------|
| Formatted: Subscript    |
| Formatted: Font: Italic |
| Formatted: Subscript    |
| Formatted: Font: Italic |
| Formatted: Subscript    |
| Formatted: Font: Italic |
| Formatted: Subscript    |
| Formatted: Font: Italic |
| Formatted: Subscript    |
| Formatted: Font: Italic |
| Formatted: Subscript    |
| Formatted: Font: Italic |
| Formatted: Subscript    |
| Formatted: Font: Italic |
| Formatted: Subscript    |

with less predictor variables. On the contrary, the  $MCC_{memd}$  also overestimates the multivariate relationships. For example, when considering only predictor variables corresponding to scales of 8, 16, and 32, the  $MCC_{memd}$  value for the stationary case is 0.47 at the scale of 64. This deviates much from the expected  $MCC_{memd}$  value of 0 (Fig. 6a). The possible underestimation and overestimation by the  $MCC_{memd}$  may come from the decomposition errors inherent in the MEMD algorithm (Rehman and Mandic, 2010).

Similar to MSC, the localized multivariate relationships cannot be obtained from MCCmemd. This can be better explained by the decrease of MCCmemd when half of the series of the predictor variables are replaced by 0 (Fig. 6b). Take the stationary case for example, the MCCmemd values at the scales corresponding to  $y_2$  and  $y_4$  decrease from 0.98 to 0.49, and from 0.93 to 0.62, respectively, when the second half of the  $y_2$ and  $y_4$  series are replaced by 0.

332 As explained above, the MWC has advantages in untangling localized multivariate 333 relationships as compared to the common multivariate methods. It is important to 334 reveal the multivariate relationships which vary with time or space, that are associated 335 with different processes. For example, discharge usually occurs on knolls, while 336 recharge usually occurs in neighboring depressions (Gates et al., 2011). Therefore, the 337 controlling factors of soil water storage may vary with the land element characteristics of a location. Local controls may be more important on knolls, while non-local 338 controls may be more important in depressions (Grayson et al., 1997). In a temporal 339 domain, vegetation transpiration contributes more to the evapotranspiration in the 340

[revised manuscript text omitted]